# Boosting Data-Driven Mirror Descent with Randomization, Equivariance, and Acceleration

**Hong Ye Tan**                                                                *hyt35@cam.ac.uk*
*Department of Applied Mathematics and Theoretical Physics*
*University of Cambridge, United Kingdom*

**Subhadip Mukherjee**                                                *smukherjee@ece.iitkgp.ac.in*
*Department of Electronics and Electrical Communication Engineering*
*Indian Institute of Technology, Kharagpur, India*

**Junqi Tang**                                                                   *j.tang.2@bham.ac.uk*
*School of Mathematics*
*University of Birmingham, United Kingdom*

**Carola-Bibiane Schönlieb**                                              *cbs31@cam.ac.uk*
*Department of Applied Mathematics and Theoretical Physics*
*University of Cambridge, United Kingdom*

**Reviewed on OpenReview:** *https://openreview.net/forum?id=r2dx1s1lqG*

## Abstract

Learning-to-optimize (L2O) is an emerging research area in large-scale optimization with applications in data science. Recently, researchers have proposed a novel L2O framework called learned mirror descent (LMD), based on the classical mirror descent (MD) algorithm with learnable mirror maps parameterized by input-convex neural networks. The LMD approach has been shown to significantly accelerate convex solvers while inheriting the convergence properties of the classical MD algorithm. This work proposes several practical extensions of the LMD algorithm, addressing its instability, scalability, and feasibility for high-dimensional problems. We first propose accelerated and stochastic variants of LMD, leveraging classical momentum-based acceleration and stochastic optimization techniques for improving the convergence rate and per-iteration computational complexity. Moreover, for the particular application of training neural networks, we derive and propose a novel and efficient parameterization for the mirror potential, exploiting the equivariant structure of the training problems to significantly reduce the dimensionality of the underlying problem. We provide theoretical convergence guarantees for our schemes under standard assumptions and demonstrate their effectiveness in various computational imaging and machine learning applications such as image inpainting, and the training of support vector machines and deep neural networks.

## 1 Introduction

Large-scale optimization plays a key role in modern data science applications. In such applications, we typically seek to infer a collection of parameters $x \in \mathbb{R}^n$ from $m$ i.i.d. data samples of a random variable $\{z_i\}_{i=1}^m$, by solving optimization problems of the form:

$$x^\star \in \operatorname*{arg\,min}_{x \in \mathbb{R}^n} f(x) := \frac{1}{m} \sum_{i=1}^m \left[ f_i(x; z_i) + R(x) \right], \tag{1}$$

where each $f_i$ is a fidelity function associated with the $i$-th data sample $z_i$, while $R(x)$ is a regularizer containing the prior knowledge of $x$. In the context of machine learning applications, $x$ would contain the parameters of some models such as SVMs and neural networks, while $z_i$ contains a pair of of training data samples and labels. For signal/image processing and inverse problems, $z_i$ is usually referred to as a pair of measurement operators and measurement data, with the measurement operator appearing in the fidelity $f_i$. Many classical variational image reconstruction models can also take the form of (1), with a simple example being total variation regularization (Rudin et al., 1992). In this case, we have $m = 1$, where $f_1$ depends only on the corrupted image, and the total variation regularizer $R(x) = \lambda \|\nabla x\|_1$ is the $\ell_1$ norm of the discrete differences of $x$. In modern applications, the dimension of the optimization problems and the number of data samples involved are often large, leading to heavy demands on the development of computationally efficient optimization algorithms.

Throughout the past decade, we have witnessed tremendous successes by researchers in the field of optimization in data science. In particular, efficient first-order algorithms utilizing momentum-based acceleration (Nesterov, 1983; Beck & Teboulle, 2009) and stochastic approximation (Robbins & Monro, 1951; Zhang, 2004) have been proposed, with applications including sparse signal recovery and convex programming (Bubeck, 2015; Ghadimi & Lan, 2016). Recently, stochastic gradient algorithms with theoretically optimal convergence rates for solving generic classes of problems were developed (Lan, 2012; Lan & Zhou, 2015; Defazio, 2016; Allen-Zhu, 2017; Chambolle et al., 2018; Tang et al., 2018; Zhou et al., 2018b; Driggs et al., 2021), matching complexity lower bounds for convex (Woodworth & Srebro, 2016) or non-convex problems (Arjevani et al., 2023).

Despite being theoretically optimal for generic optimization problem classes, stochastic gradient methods result in poor performance when applied to problems with particular structures. Tang et al. (2020) demonstrate that while optimal stochastic gradient methods perform very well in terms of convergence rates for certain imaging inverse problems such as X-ray CT, they perform poorly for some other problems such as image deblurring and compressed sensing. One important aspect is the fact that these methods are designed for some wide generic class of problems, while in practice, each practitioner may be only interested in a very narrow subclass of problems with a specific structure. The optimal algorithms for generic problems may still perform suboptimally for the specific subclasses of interest.

## 1.1 Learning-to-Optimize

Learning-to-optimize (L2O), sometimes referred to as meta-learning in the machine learning literature, aims to overcome this limitation when dealing with more specialized problem classes, such as natural image reconstruction as a subset of all imaging problems. An underlying idea is to utilize the intrinsic structure of the optimization problems to converge faster on tasks from the same distribution. This has been shown to be effective for problems such as linear regression, training neural networks, and tomographic reconstruction, beating standard analytic methods (Andrychowicz et al., 2016; Li & Malik, 2016; Li et al., 2019; Banert et al., 2020). The aim is to generate optimizers that (1) optimize in-distribution functions at a faster rate, and (2) possibly return a higher quality solution given the same computing budget (Chen et al., 2022).

The L2O framework can be formally described as follows (Chen et al., 2022). Given a learnable mapping function $g$ with parameters $\theta$, as well as a set $\mathcal{F}$ of training sample optimization problems $\min_x f(x)$ for $f \in \mathcal{F}$, the optimizer is learned to generate updates of the form $x_{k+1} = x_t - g_\theta(I_t)$. Here, $I_t$ represents previous information, such as the iterates $x_t, x_{t-1}, ...$ or their gradients $\nabla f(x_t), \nabla f(x_{t-1})$ etc. For example, gradient descent with can be formulated as $g_\theta(I_t) = \eta \nabla f(x_t)$, with a learnable step-size parameter $\eta$. This formulation of L2O is flexible in the sense that all previous information is available, which should theoretically allow for better informed steps. The parameters are usually learned by minimizing an objective of the form

$$\min_\theta \mathbb{E}_{f \in \mathcal{F}} \left[ \sum_{i=1}^{T} w_t f(x_t) \right], \quad x_{t+1} = x_t - g_\theta(I_t). \tag{2}$$

This can be interpreted as minimizing the objective value of the evolved iterates, with weights $w_t$. Some standard choices include $w_t \equiv 1$, or $w_T = 1, w_{T-1} = ... = w_1 = 0$ (Zhou et al., 2018a; Andrychowicz et al., 2016).

## 1.2 Existing Theory in L2O

Learning-to-optimize frameworks broadly fall into two categories: fully learned "model-free" methods, and "model-based" methods based on classical algorithms (Chen et al., 2022). Model-free methods directly parameterize the optimizer updates $g_\theta(I_t)$ using neural networks, formulating fast optimization as an unrolled loss objectives of the form Equation (2) (Andrychowicz et al., 2016; Li & Malik, 2016). Recent advances for this paradigm include improving the generalization performance of both the optimizers and final optimization objective when used to train neural networks (Almeida et al., 2021; Yang et al., 2023), more stable computation (Metz et al., 2019), and learning to warm-start certain optimization methods (Sambharya et al., 2023). However, major drawbacks include lack of convergence guarantees, as well as requiring many training samples.

Model-based L2O methods instead inherit structure from classical optimization algorithms such as PDHG-like proximal splitting algorithms, where certain optimizer parameters such as step-sizes or momentum parameters are learned (Banert et al., 2020; Almeida et al., 2021; Banert et al., 2021; Gregor & LeCun, 2010). These methods are typically applied to convex problems rather than non-convex problems such as neural network training, which also allows for theoretical convergence analysis of the learned schemes. One advantage of building a L2O model from a classical optimization model is that convergence of the learned model can usually be derived by slightly modifying existing convergence results. A more in-depth overview of L2O can be found in Chen et al. (2022).

From a more statistical point of view, Chen & Hazan (2023) formulates learning-to-optimize as a feedback control problem for smooth convex objective functions, showing an upper bound on the expected optimality gap in a broader class of stabilizing controllers. In the case where the algorithm class has small pseudo-dimension with respect to a set of problem instances, the empirical risk minimization problem admits statistical PAC bounds for the number of problem instances required (Gupta & Roughgarden, 2016). Sambharya et al. (2023) demonstrates PAC-Bayes generalization bounds for certain classes of fixed-point optimization methods. A related notion is that of (semi-)amortized optimization, where a model maps an initialization to a solution with access to an objective $f$ (Amos et al., 2023).

Combining L2O with mirror descent has mainly been explored in the online setting, with theoretical contributions arising in terms of statistical transfer bounds and optimality gap bounds (Khodak et al., 2019; Denevi et al., 2019). Gao et al. (2022) also consider a simple version of coordinate-based mirror descent, providing a statistical PAC generalization bound. These are more suitable to common meta-learning benchmarks such as few-shot training for classification where convexity is unavailable (Finn et al., 2017; Nichol et al., 2018). These meta-learning methods that are focused on neural networks generally take a different approach to the L2O methods discussed above – meta-learning accelerates by moving the initializations to "better points", in contrast to the L2O objective of learning an optimization trajectory (from any initialization). We consider a more general and less theoretically-backed framework where the meta-parameters are baked into the optimizer, rather than common instances such as initialization which are difficult to interpret in our context.

A provable learning-to-optimize framework called learned mirror descent (LMD) was recently proposed in Tan et al. (2023b;a) based on the classical mirror descent (MD) algorithm by Nemirovski & Yudin (1983). By using neural networks to implicitly learn and model the geometry of the underlying problem class via mirror maps, LMD can achieve significantly faster convergence on certain convex problem classes, including model-based image inpainting and training support vector machines. Moreover, the LMD framework proposes approximate convergence results based on the accuracy of the mirror map inversion.

Building upon the vanilla LMD framework by Tan et al. (2023b), we make the following contributions:

1. **Learned accelerated mirror descent and its theoretical analysis.** Our first contribution is developing the learned accelerated mirror descent (LAMD) scheme, which was initially proposed in the preliminary work of Tan et al. (2023a) without convergence analysis. In this work, we complete the theoretical analysis of the LAMD scheme under standard assumptions, demonstrating improved convergence rates over vanilla LMD. In particular, we demonstrate in Theorem 1 that we obtain convergence to the minimum, as opposed to a constant above the minimum as found in Tan et al. (2023b).

2. **Learned stochastic mirror descent and its acceleration.** We propose the learned stochastic mirror descent (LSMD) and the accelerated version (LASMD) in the spirit of stochastic optimization, which is crucial for the scalability of the LMD framework in large-scale problems prevalent in modern data science applications. We also provide theoretical convergence analysis of these schemes and demonstrate their effectiveness using numerical experiments including image processing, training of SVMs, and training neural networks for classification.

3. **Efficient mirror map parameterization by utilizing the equivariant structure.** We propose equivariant theory for L2O in the case where the target class of objectives satisfies some group symmetries. For the training of deep neural networks, we utilize the equivariant structure of the training problems to significantly simplify the parameterization of the mirror map, further improving the efficiency and practicality of the LMD framework, and allowing for applications in a non-convex setting. We provide a derivation of this scheme, showing theoretically that learned optimizers trained under certain group invariance and equivariance assumptions will also satisfy a group invariance property, removing redundant parameters and reducing the problem dimensionality. We additionally utilize this to extend the LMD framework to the task of training deep and convolutional neural networks, achieving competitive performance with widely used and empirically powerful optimizers such as Adam.

### 1.3 Mirror Descent and Learned Mirror Descent

Mirror descent (MD) was originally proposed by Nemirovski & Yudin (1983) as a method of generalizing gradient descent to general infinite-dimensional Banach spaces. The lack of Hilbert space structure and isomorphisms between $\mathcal{X}$ and their duals $\mathcal{X}^*$ prevents the use of gradient descent, which usually identifies the formal derivative $Df \in \mathcal{X}^*$ with the corresponding Riesz element $\nabla f \in \mathcal{X}$, satisfying $\langle Df, x \rangle_{\mathcal{X}^*, \mathcal{X}} = \langle \nabla f, x \rangle_{\mathcal{X}, \mathcal{X}}$. MD proposes to address this by directly updating elements in the dual space $\mathcal{X}^*$, and 'tying' the dual space updates in $\mathcal{X}^*$ to corresponding primal space updates in $\mathcal{X}$ using a "mirror map", satisfying various conditions.

Generalizing the classical Euclidean structure used for gradient descent to more general Bregman divergences, such as those induced by $L^p$ norms, can improve the dimensionality scaling of the Lipschitz constant. Having a smaller Lipschitz constant with respect to another norm compared to the Euclidean norm directly impacts the step size for which the methods converge, as well as the convergence rates. For example, for minimizing a convex function $f$ on the $n$-simplex $\Delta^n = \{x \in \mathbb{R}^n \mid x_i \geq 0, \sum_{i=1}^n x_i\}$, MD is able to achieve rates of $\mathcal{O}(\sqrt{\log n} \|f'\|_\infty / \sqrt{k})$, as opposed to (sub-)gradient descent which has rate $\mathcal{O}(\|f'\|_2 / \sqrt{k})$ (Beck & Teboulle, 2003; Ben-Tal et al., 2001). The ratio $\|f'\|_2 / \|f'\|_\infty$ can be as large as $\sqrt{n}$, leading to $(n/\log n)^{1/2}$ asymptotic speedup by using MD.

Gunasekar et al. (2021) demonstrate further that mirror descent corresponds to a Riemannian gradient flow in the limiting case, and further generalize mirror descent to general Riemannian geometries. Moreover, the mirror descent framework is amenable to modifications similar to gradient descent, with extensions such as ergodicity (Duchi et al., 2012), composite optimization (Duchi et al., 2010), stochasticity (Lan et al., 2012; Zhou et al., 2017) and acceleration (Krichene et al., 2015; Hovhannisyan et al., 2016). In summary, the mirror descent algorithm allows us to utilize non-Euclidean geometry for optimization, which has been shown to accelerate convergence in applications including online learning and tomography (Srebro et al., 2011; Raskutti & Mukherjee, 2015; Allen-Zhu & Orecchia, 2014; Ben-Tal et al., 2001; Orabona et al., 2015; Zimmert & Lattimore, 2019).

We present the MD algorithm as given in Beck & Teboulle (2003), as well as the data-driven version of which this work is based, LMD (Tan et al., 2023b). The LMD framework serves as a base for the accelerated and stochastic extensions that will be presented in this work. LMD aims to speed up convergence on a class of "similar" functions, which are taken to be qualitatively similar, such as image denoising on natural images or CT imaging (Banert et al., 2020). By using data to generate functions from the target function class, the geometry of the class can be learned using these sample functions. Mirror descent then exploits this learned geometry for faster convergence. Follow-up works additionally suggest that the learned geometry can be transferred to other mirror-descent-type algorithms without the need for retraining. Moreover, the learned

mirror maps are robust to small perturbations in the forward operator such as from the identity to a blur convolution, as well as change of data sets (Tan et al., 2023a).

We begin with some notation and definitions that will be used for mirror descent, as well as more standard assumptions for the MD framework.

Let $\mathcal{X}$ be the finite dimensional normed space $(\mathbb{R}^n, \|\cdot\|)$. Denote by $\mathcal{X}^* = ((\mathbb{R}^n)^*, \|\cdot\|_*)$ the dual space, where the dual norm is defined as $\|p\|_* = \sup_x\{\langle p, x\rangle \mid \|x\| \leq 1\}$, and the bracket notation denotes evaluation $\langle p, x\rangle = p(x) \in \mathbb{R}$. Let $\overline{\mathbb{R}} := \mathbb{R} \cup \{+\infty\}$ denote the extended real line. Let $f : \mathcal{X} \to \overline{\mathbb{R}}$ be a proper, convex, and lower-semicontinuous function with well-defined subgradients. For a differentiable convex function $h : \mathcal{X} \to \mathbb{R}$, define the *Bregman divergence* as $B_h(x, y) = h(x) - h(y) - \langle h'(y), x - y\rangle$. Let $g : \mathcal{X} \to \mathcal{X}^*$ be a function such that $g(x) \in \partial f(x)$ for all $x \in \mathcal{X}$. We denote a (possibly noisy) stochastic approximation to $g$ by

$$\mathsf{G}(x, \xi) = g(x) + \Delta(x, \xi) + U(x),$$

where $\mathbb{E}_\xi[\Delta(x, \xi)] = 0$, and $U(x) : \mathcal{X} \to \mathcal{X}^*$ represents dual noise. One standard assumption that we will use is that $\mathsf{G}$ has bounded second moments.

While we have defined $\mathcal{X}$ to be a vector space, this is not necessary. Indeed, suppose instead that $\mathcal{X}$ were a (proper) closed convex subset of $\mathbb{R}^n$. Assuming that the inverse mirror function maps into $\mathcal{X}$, i.e., $\nabla\psi^* : (\mathbb{R}^n)^* \to \mathcal{X}$, such as when $\nabla\psi(x) \to \infty$ as $x \to \partial\mathcal{X}$, then the proposed algorithms will still hold. We will restrict our exposition in this work to the simpler case where $\mathcal{X} = \mathbb{R}^n$ to avoid these technicalities, but the results can be extended.

**Definition 1.** *We define a* mirror potential *to be an $\alpha$-strongly convex $\mathcal{C}^1$ function $\psi : \mathcal{X} \to \mathbb{R}$ with $\alpha > 0$. We define a* mirror map *to be the gradient of a mirror potential $\nabla\psi : \mathcal{X} \to \mathcal{X}^*$.*

We further denote the convex conjugate of $\psi$ by $\psi^* : \mathcal{X}^* \to \mathbb{R}$. Note that under these assumptions, we have that $\nabla\psi^* = (\nabla\psi)^*$ on $\operatorname{dom}\psi$. Moreover, $\psi^*$ is $\alpha^{-1}$-smooth with respect to the dual norm, or equivalently, $\nabla\psi^*$ is $\alpha^{-1}$-Lipschitz (Azé & Penot, 1995).

For a mirror potential $\psi$ and step-sizes $(t_k)_{k=0}^\infty$, the mirror descent iterations of Beck & Teboulle (2003) applied to an initialization $x^{(0)} \in \mathcal{X}$ are

$$x^{(k+1)} = \nabla\psi^*(\nabla\psi(x^{(k)}) - t_k\nabla f(x^{(k)})). \tag{3}$$

The roles of $\nabla\psi : \mathcal{X} \to \mathcal{X}^*$ and $\nabla\psi^* : \mathcal{X}^* \to \mathcal{X}$ are to *mirror* between the primal and dual spaces such that the gradient step is taken in the dual space. A special case is when $\Psi(x) = \frac{1}{2}\|x\|_2^2$, whereby $\nabla\psi$ and $\nabla\psi^*$ are the canonical isomorphisms between $\mathcal{X}$ and $\mathcal{X}^*$, and the MD iterations (3) reduce to gradient descent. In the case where $f$ is $L$-Lipschitz and $\psi$ has strong convexity parameter $\alpha > 0$, mirror descent is able to achieve optimality gap bounds of the following form (Beck & Teboulle, 2003):

$$\min_{1 \leq k \leq s} f(x^{(k)}) - f(x^*) \leq \frac{B_\psi(x^*, x^{(1)}) + (2\alpha)^{-1}\sum_{k=1}^s t_k^2\|\nabla f(x^{(k)})\|_*^2}{\sum_{k=1}^s t_k}.$$

LMD arises when we replace $\psi$ and $\psi^*$ with neural networks, which we can then learn from data. We denote learned variants of $\psi$ and $\psi^*$ by $M_\theta : \mathcal{X} \to \mathbb{R}$ and $M_\vartheta^* : \mathcal{X}^* \to \mathbb{R}$ respectively. Note that $M_\theta$ and $M_\vartheta^*$ are not necessarily convex conjugates of each other due to the lack of a closed-form convex conjugate for general neural networks – we change the subscript to remind the reader of this subtlety. Equipped with this new notation for learned mirror maps, the learned mirror descent algorithm arises by directly replacing the mirror maps $\nabla\psi$ and $\nabla\psi^*$ with learned variants $M_\theta$ and $M_\vartheta^*$ respectively, where $\nabla M_\vartheta^*$ is enforced to be approximately $(\nabla M_\theta)^{-1}$ during training (Tan et al., 2023b):

$$x^{(k+1)} = \nabla M_\vartheta^*(\nabla M_\theta(x^{(k)}) - t_k\nabla f(x^{(k)})). \tag{4}$$

The mismatch between $\nabla M_\vartheta^*$ and $(\nabla M_\theta)^{-1}$, also called the *forward-backward inconsistency*, necessarily arises as the convex conjugate of a neural network is generally intractable. Tan et al. (2023b) consider analyzing LMD as an approximate mirror descent scheme, where the error bounds depend on the distance between

the computed (approximate) iterates (4) and the true mirror descent iterates (3). Under certain conditions on the target function $f$ and the mirror potential $\psi = M_\theta$, they show convergence in function value up to a constant over the minimum (Tan et al., 2023b, Thm. 3.1, 3.6). A sufficient condition for this convergence is that the LMD iterations $x_i$ with inexact mirror maps given by Equation (4) is close to the exact MD iterations $\hat{x}^{(k+1)} = (\nabla M_\theta)^{-1}(\nabla M_\theta(x^{(k)}) - t_k \nabla f(x^{(k)}))$, in the sense that $\langle \nabla \psi(\hat{x}^{(i)}) - \nabla \psi(x^{(i)}), x - x^{(i)} \rangle$ and $\langle \nabla f(\hat{x}_i), x_i - \hat{x}_i \rangle$ are uniformly bounded for a given minimizer $x$.

This work proposes to address two current restrictions of LMD. The first restriction is the approximate convergence and instability that arises due to the forward-backward inconsistency, such that the function values are only minimized up to a constant above the minimum. This is demonstrated in Tan et al. (2023a, Thm. 3.1), where LMD is also shown to diverge in later iterations as the forward-backward inconsistencies accumulate. Section 2 presents an algorithm through which the constant can vanish if the forward-backward inconsistency is uniformly bounded. The second drawback is the computational cost of training LMD, which is prohibitive for expensive forward operators such as CT or neural network parameters. Sections 3 and 4 present stochastic and accelerated stochastic extensions of LMD that show convergence in expectation even with possibly unbounded stochastic approximation errors. We demonstrate these extensions with function classes considered in previous works in Section 5, utilizing pre-trained neural networks on tasks such as image denoising, image inpainting and training support vector machines (Tan et al., 2023b;a).

Section 6 introduces a method of reducing the dimensionality of the mirror maps without affecting the expressivity, by exploiting the symmetries in the considered functions, with applications to training mirror maps on neural networks. Moreover, the lower dimensionality allows us to consider parameterizations of mirror maps with exact inverses, bypassing the forward-backward inconsistency problem. We utilize equivariance to extend a simple version of LMD to train deep neural networks in Section 7, where we demonstrate competitive performance with Adam on non-convex problem classes.

## 2   Learned Accelerated MD

In this section, we first present a mirror descent analog to the Nesterov accelerated gradient descent scheme (Nesterov, 1983). The resulting scheme, named accelerated mirror descent (AMD), can be considered as a discretization of a coupled ODE. Krichene et al. (2015) show accelerated convergence rates of the ODE, which translate to a discrete-time algorithm after a particular discretization. The analysis and convergence rates of AMD are similar to the ODE formulation for Nesterov accelerated gradient descent given in Su et al. (2014), where both papers show $\mathcal{O}(1/k^2)$ convergence rate of function value.

By considering the discrete AMD iterations with inexact mirror maps, we then generalize this algorithm to the approximate case, given in Algorithm 1, and provide the corresponding convergence rate bound. This can then be applied in the case where the forward and backward mirror maps are given by neural networks, leading to convergence results for learned accelerated mirror descent (LAMD). Our main contribution in this section is Theorem 1, which shows convergence of the function value to the minimum, rather than to the minimum plus a constant.

---

**Algorithm 1** Accelerated mirror descent (AMD) with approximate mirror updates

---

**Require:** $\tilde{x}^{(0)} = \tilde{z}^{(0)} = x^{(0)}$, step-sizes $t_k$, parameter $r \geq 3$
1: **for** $k \in \mathbb{N}$ **do**
2: $\quad x^{(k+1)} = \lambda_k \tilde{z}^{(k)} + (1 - \lambda_k)\tilde{x}^{(k)}$ with $\lambda_k = \frac{r}{r+k}$
3: $\quad \tilde{z}^{(k+1)} = \nabla M_{\tilde{\vartheta}}^*(\nabla M_\theta(\tilde{z}^{(k)}) - \frac{k t_k}{r}\nabla f(x^{(k+1)}))$
4: $\quad \tilde{x}^{(k+1)} = \arg\min_{\tilde{x} \in \mathbb{R}^n} \gamma t_k \left\langle \nabla f(x^{(k+1)}), \tilde{x} \right\rangle + R(\tilde{x}, x^{(k+1)})$
5: **end for**

---

Step 3 of Algorithm 1 is the mirror descent step of AMD, where $\tilde{z}^{(k)}$ represents a (computable) approximate mirror descent iteration. The additional step in Step 4 arises as a correction term to allow for convergence analysis (Krichene et al., 2015). Here, $R$ is a regularization function where there exists $0 < \ell_R \leq L_R$ such

that for all $x, x' \in \mathcal{X}$, we have

$$\frac{\ell_R}{2}\|x - x'\|^2 \le R(x, x') \le \frac{L_R}{2}\|x - x'\|^2.$$

For practical purposes, $R$ can be taken to be the Euclidean distance $R(x, x') = \frac{1}{2}\|x - x'\|^2$, in which case $\ell_R = L_R = 1$. Taking $R$ of this form reduces Step 4 to the following gradient descent step:

$$\tilde{x}^{(k+1)} = x^{(k+1)} - \gamma t_k \nabla f(x^{(k+1)}).$$

Similarly to Tan et al. (2023b), we additionally define a sequence $\hat{z}^{(k)}$ that comprises the true mirror descent updates, applied to the intermediate iterates $\tilde{z}^{(k)}$:

$$\hat{z}^{(k+1)} = (\nabla M_\theta)^{-1}\left(\nabla M_\theta(\tilde{z}^{(k)}) - \frac{kt_k}{r}\nabla f(x^{(k+1)})\right). \tag{5}$$

This additional variable will allow us to quantify the approximation error, with which we will show the approximate convergence. For ease of notation, we will denote the forward mirror potential by $\psi = M_\theta$. With this notation, $\psi^*$ is the convex conjugate, satisfying $\nabla\psi^* = (\nabla\psi)^{-1}$. The true mirror descent auxiliary sequence can be written as $\hat{z}^{(k+1)} = \nabla\psi^*\left(\nabla\psi(\tilde{z}^{(k)}) - \frac{kt_k}{r}\nabla f(x^{(k+1)})\right)$. We begin with the following lemma, which bounds the difference between consecutive energies to be the sum of a negative term and an approximation error term.

**Lemma 1.** *Consider the approximate AMD iterations from Algorithm 1, and let $\hat{z}^{(k)}$ be the exact MD iterates given by Equation 5. Assume $\psi^*$ is $L_{\psi^*}$-smooth with respect to a reference norm $\|\cdot\|_*$ on the dual space, i.e. $B_{\psi^*}(z, y) \le \frac{L_{\psi^*}}{2}\|z - y\|_*^2$, or equivalently that $\nabla\psi^*$ is $L_{\psi^*}$-Lipschitz. Assume further that there exists $0 < \ell_R \le L_R$ such that for all $x, x' \in \mathcal{X}$, $\frac{\ell_R}{2}\|x - x'\|^2 \le R(x, x') \le \frac{L_R}{2}\|x - x'\|^2$. Define the energy $\tilde{E}^{(k)}$ for $k \ge 0$ as follows (where $t_{-1} = 0$):*

$$\tilde{E}^{(k)} := \frac{k^2 t_{k-1}}{r}\left(f(\tilde{x}^{(k)}) - f^*\right) + rB_{\psi^*}\left(\nabla\psi(\tilde{z}^{(k)}), \nabla\psi(x^*)\right). \tag{6}$$

*Assume the step-size conditions $\gamma \ge L_R L_{\psi^*}$, and $t_k \le \frac{l_R}{L_f \gamma}$. Then the difference between consecutive energies satisfies the following:*

$$\tilde{E}^{(k+1)} - \tilde{E}^{(k)} \le \frac{(2k+1-rk)t_k}{r}\left(f(\tilde{x}^{(k+1)}) - f^*\right) - \frac{k^2(t_{k-1} - t_k)}{r}\left(f(\tilde{x}^{(k)}) - f^*\right)$$
$$+ r\left\langle\nabla\psi(\tilde{z}^{(k+1)}) - \nabla\psi(\hat{z}^{(k+1)}), \tilde{z}^{(k+1)} - x^*\right\rangle.$$

*Proof.* Deferred to Appendix A.1. $\qquad\square$

Lemma 1 gives us a way of telescoping the energy $\tilde{E}$. To turn this into a bound on the objective values $f(\tilde{x}^{(k)}) - f^*$, we need the initial energy. The following proposition gives a bound on the energy $\tilde{E}^{(1)}$, which when combined with telescoping with Lemma 1, will be used to derive bounds on the objective.

**Proposition 1.** *Assume the conditions as in Lemma 1. Then*

$$\tilde{E}^{(1)} \le rB_{\psi^*}(\nabla\psi(x^{(0)}), \nabla\psi(x^*)) + \frac{t_0}{r}(f(x^{(0)}) - f^*) + r\left\langle\nabla\psi(\tilde{z}^{(1)}) - \nabla\psi(\hat{z}^{(1)}), \tilde{z}^{(1)} - x^*\right\rangle \tag{7}$$

*Proof.* The proof relies on bounding $f(\tilde{x}^{(1)}) - f^*$, which is done identically to that in Krichene et al. (2015). Deferred to Appendix A.2. $\qquad\square$

Putting together Lemma 1 and Proposition 1, we get the following theorem that bounds the objective value of the approximate AMD iterates.

**Theorem 1.** *Assume the conditions as in Lemma 1. Assume also that the step-sizes $(t_k)_{k=0}^\infty$ are non-increasing, and that the approximation error term, given by*

$$\left\langle \nabla\psi(\tilde{z}^{(k+1)}) - \nabla\psi(\hat{z}^{(k+1)}), \tilde{z}^{(k+1)} - x^* \right\rangle, \tag{8}$$

*is uniformly bounded by a constant $M > 0$. If our step-sizes are chosen as $t_k = \Theta(k^{-c})$ for $c \in [0,1)$, then we get $\mathcal{O}(k^{c-1})$ convergence in objective value.*

*Proof.* Summing the expression in Lemma 1, noting the first two terms are non-positive for $k \geq 1$ since $t_k$ is non-increasing and $r \geq 3$, and noting that the final term is bounded by $rM$,

$$\tilde{E}^{(k+1)} \leq \tilde{E}^{(1)} + krM.$$

By definition of $\tilde{E}$ and non-negativity of $B_{\psi^*}$, we also have

$$f(\tilde{x}^{(k+1)}) - f^* \leq \frac{r}{(k+1)^2 t_k}\tilde{E}^{(k+1)} \leq \frac{r}{(k+1)^2 t_k}[\tilde{E}^{(1)} + krM] = \mathcal{O}(k^{c-1}). \tag{9}$$

$\square$

**Remark 1.** *The condition that Equation (8) is bounded can be interpreted in terms of more natural functional bound on the mirror potentials $M_\theta, M_\vartheta^*$. Indeed, the LHS is simply $(\nabla M_\theta \circ \nabla M_\vartheta^* - I)(\nabla M_\theta(\tilde{z}^{(k)}) - (kt_k/r)\nabla f(x^{(k+1)}))$. Therefore, if $\|\nabla M_\theta \circ \nabla M_\vartheta^* - I\|$ is uniformly bounded (over the iterates in the dual space) and the iterates are bounded, by Cauchy-Schwarz, the approximation error term Equation (8) will also be bounded.*

This theorem shows that we are actually able to obtain convergence to the function objective minimum by using acceleration, as opposed to the minimum plus a constant given in Tan et al. (2023b). There are two main ideas at play here. One is that the acceleration makes the function value decrease fast enough, as seen by the energy given in Equation 6 growing as $\mathcal{O}(k)$, while having a $k^2 t_k$ factor in front of the objective difference term. The second idea is that the $\lambda_k$ factor in Step 2 of Algorithm 1 decreases the importance of the mirror descent step, which thus reduces the effect of the forward-backward inconsistency.

## 3 Learned Stochastic MD

Stochastic mirror descent (SMD) arises when we are able to sample gradients $\mathsf{G}(x^k, \xi^k)$ such that the expectation of $\mathsf{G}$ is equal to a subgradient of our convex objective $f$. For example, in large-scale imaging applications such as CT, computing the forward operator may be expensive, and a stochastic gradient may consist of computing a subset of the forward operator. Another example would be neural network training over large data sets, where there is insufficient memory to keep track of the gradients of the network parameters over all of the given data. Therefore, we wish to extend the analysis for approximate mirror descent to the case where we only have access to stochastic gradients.

While it may be tempting to put the stochastic error into the approximation error terms of the previous analyses, stochastic errors may be unbounded, violating the theorem assumptions. This section will extend approximate MD to the case where the approximation error is now the sum of a bounded term and a zero-mean term with finite second moments. In particular, Theorem 2 shows an expected optimality gap bound depending on the stochastic noise based on bounded variance and bounded error assumptions.

We base the analysis of this section on the robust stochastic approximation approach (Nemirovski et al., 2009). We require two additional assumptions in this setting as follows:

**Assumption 1.** *We can draw i.i.d. samples $\xi^{(0)}, \xi^{(1)}, \dots$ of a random vector $\xi \in \Xi$.*

**Assumption 2.** *An oracle $\mathsf{G}$ exists for which, given an input $(x, \xi) \in \mathcal{X} \times \Xi$, returns a vector $\mathsf{G}(x, \xi)$ such that $g(x) := \mathbb{E}_\xi[\mathsf{G}(x, \xi)]$ is well defined and satisfies $g(x) \in \partial f(x)$.*

If $f$ can be written as an expectation $f(x) = \mathbb{E}[\mathsf{F}(x,\xi)]$, where $\mathsf{F}(\cdot,\xi)$ is convex with $f$ having finite values in a neighborhood of a point $x$, then we can interchange the subgradient with the expectation (Strassen, 1965),

$$\partial f(x) = \mathbb{E}[\partial_x \mathsf{F}(x,\xi)].$$

Stochastic MD thus generalizes MD by replacing the subgradient $\nabla f$ at each step with this gradient oracle $\mathsf{G}$, which can be written as follows

$$x^{(k+1)} = \nabla\psi^* \left( \nabla\psi(x^{(k)}) - t_k \mathsf{G}(x^{(k)}, \xi^{(k)}) \right). \tag{10}$$

Observe that under this formulation, we can additionally encode a noise component that arises as a result of inexact mirror descent computation. Therefore, we may redefine $\mathsf{G}(x,\xi)$ as an inexact stochastic oracle as in the introduction, having two components

$$\mathsf{G}(x,\xi) = g(x) + \Delta(x,\xi) + U(x),$$

where $\Delta(\cdot,\xi)$ signifies stochastic noise satisfying $\mathbb{E}[\Delta(x)] = \mathbf{0}$, and $U(x)$ is a deterministic error to model the approximation error of computing MD steps. We will use the notation $\Delta_k = \Delta(x^{(k)}, \xi^{(k)})$, $U_k = U(x^{(k)})$ to signify these values at each iteration.

---

**Algorithm 2** Stochastic mirror descent (SMD) (Nemirovski et al., 2009)

---

**Require:** $x^{(0)} \in \mathcal{X}$, step-sizes $t_k > 0$, stochastic oracle $\mathsf{G}$, random i.i.d. $\xi^{(k)}$
1: **for** $k \in \mathbb{N}$ **do**
2:     $x^{(k+1)} = \nabla\psi^*(\nabla\psi(x^{(k)}) - t_k \mathsf{G}(x^{(k)}, \xi^{(k)}))$
3: **end for**

---

We first reformulate the MD iterates into a "generalized proximal" form. In particular, a small modification of the argument in Beck & Teboulle (2003) shows that the SMD iterations given in Equation (10) can be written as follows:

$$x^{(k+1)} = \operatorname*{arg\,min}_{x' \in \mathcal{X}} \left\{ \left\langle \mathsf{G}(x^{(k)}, \xi^{(k)}), x' \right\rangle + \frac{1}{t_k} B_\psi(x', x^{(k)}) \right\}. \tag{11}$$

This can be written in terms of the *prox-mapping* $P_x : (\mathbb{R}^n)^* \to \mathcal{X}$, defined as follows:

$$P_x(y) = \operatorname*{arg\,min}_{x' \in \mathcal{X}} \left\{ \langle y, x' \rangle + B_\psi(x', x) \right\}, \tag{12a}$$

$$x^{(k+1)} = P_{x^{(k)}} \left( t_k \mathsf{G}(x^{(k)}, \xi^{(k)}) \right). \tag{12b}$$

The following result gives optimality gap bounds of approximate SMD in the deterministic-plus-stochastic noise setting. In particular, the expected ergodic average is able to attain a loss that is a constant over the minimum.

**Theorem 2.** *Consider the approximate SMD iterations $x^{(i)}$ generated by Equation 10. Suppose that the stochastic oracle satisfies $\mathbb{E}[\|\mathsf{G}(x,\xi)\|_*^2] \leq \sigma^2$ for all $x \in \mathcal{X}$ for some $\sigma \geq 0$. Let $x^*$ be some minimizer of $f$.*

   *1. If $\langle U_i, x^{(i)} - x^* \rangle$ is uniformly bounded by $C$, then the expected loss satisfies*

$$\mathbb{E}\left[ \sum_{i=0}^{k} t_i [f(x^{(i)}) - f^*] \right] \leq B_\psi(x^*, x^{(0)}) + \frac{\sigma^2}{2\alpha} \sum_{i=0}^{k} t_i^2 + C \sum_{i=0}^{k} t_i. \tag{13}$$

   *2. If $f$ is $\mu$-strongly convex and $\|U_i\|_*^2$ is uniformly bounded by $C'$, the expected loss satisfies*

$$\mathbb{E}\left[ \sum_{i=0}^{k} t_i [f(x^{(i)}) - f^*] \right] \leq B_\psi(x^*, x^{(0)}) + \frac{\sigma^2}{2\alpha} \sum_{i=0}^{k} t_i^2 + \frac{C'}{2\mu} \sum_{i=0}^{k} t_i. \tag{14}$$

*In particular, the ergodic average defined by*

$$\tilde{x}_0^k = \sum_{i=0}^{k} \gamma_i x^{(i)}, \quad \gamma_i = \frac{t_i}{\sum_{i=0}^{k} t_i}$$

*satisfies respectively*

$$\mathbb{E}\left[f(\tilde{x}_0^k) - f^*\right] \leq \left(B_\psi(x^*, x^{(0)}) + \frac{\sigma^2}{2\alpha} \sum_{i=0}^{k} t_i^2\right) \Big/ \left(\sum_{i=0}^{k} t_i\right) + C, \tag{15}$$

$$\mathbb{E}\left[f(\tilde{x}_0^k) - f^*\right] \leq \left(B_\psi(x^*, x^{(0)}) + \frac{\sigma^2}{2\alpha} \sum_{i=0}^{k} t_i^2\right) \Big/ \left(\sum_{i=0}^{k} t_i\right) + \frac{C'}{2\mu}. \tag{16}$$

*Proof.* Deferred to Appendix B. □

**Remark 2.** *A stronger assumption to condition (1) in Theorem 2 is to assume uniformly bounded iterates, as well as uniformly bounded deterministic errors $U_i$. In LMD, deterministic errors arise due to mismatches in the learned mirror maps $\nabla M_\vartheta^* \approx (\nabla M_\theta)^{-1}$, which can be controlled using a soft penalty.*

We observe in Equations (15) and (16) that $\ell_2$–$\ell_1$ summability (where $\sum t_i^2 < +\infty$ but $\sum t_i = +\infty$) is a sufficient condition to remove the contribution of the first term to the ergodic average error. This is consistent with the empirical observation in Tan et al. (2023a) that extending the learned step-sizes using a reciprocal rule $t_k = c/k$ gives the best convergence results.

## 4 Learned Accelerated Stochastic MD

In the previous sections, we considered accelerated and stochastic variants of mirror descent and presented results pertaining to the convergence of these algorithms with noisy mirror maps. These two variants improve different parts of MD, with acceleration improving the convergence rates, while stochasticity improves the computational dependency on the gradient. Indeed, one of the drawbacks of SMD is the slow convergence rate of $\mathcal{O}(1/\sqrt{k})$ in expectation, where the constant depends on the Lipschitz constant of $f$. Acceleration as a tool to counteract this decrease in convergence rate for SMD has recently been explored, with convergence results such as convergence in high probability (Ene & Nguyen, 2022; Lan, 2020) and in expected function value (Xu et al., 2018; Lan, 2012). These approaches decouple the gradient and stochastic noise, resulting in $\mathcal{O}(L/k^2 + \sigma/\sqrt{k})$ convergence rate, where $C$ depends on the Lipschitz constant of $\nabla f$, and $\sigma^2$ is the variance of the stochastic gradient.

We note that it is possible to extend Algorithm 1 (approximate AMD) to the stochastic case by directly replacing the gradient $\nabla f$ with a stochastic oracle $\mathsf{G}$, albeit currently without convergence guarantees. In this section, we consider another version of approximate accelerated SMD that comes with convergence guarantees, using a slightly different proof technique.

For our analysis, we follow the accelerated SMD setup in Xu et al. (2018), in particular of Algorithm 2, replicated as below. Suppose that $f$ is differentiable and has $L_f$-Lipschitz gradient, and let $\mathsf{G}(x, \xi) = \nabla_x \mathsf{F}(x, \xi)$ be a stochastic gradient satisfying

$$\mathbb{E}\left[\mathsf{G}(x^{(k)}, \xi^{(k)})\right] = \nabla f(x^{(k)}), \ \mathbb{E}\left[\|\mathsf{G}\|_*^2 \mid x^{(k)}\right] \leq \sigma^2.$$

For simplicity, we let $\Delta_k = \mathsf{G}(x^{(k)}, \xi^{(k)}) - \nabla f(x^{(k)})$ denote the zero mean stochastic component.

An inexact convex conjugate can be modeled using a noise term in Step 3. With a noise term $U$, the step would instead read

$$x^{(k+1)} = \frac{\tau_k}{\tau_k + 1}\left[\nabla \psi^*(y^{(k)}) + U(y^{(k)})\right] + \frac{1}{\tau_k + 1} x^{(k)}. \tag{17}$$

---

**Algorithm 3** Accelerated stochastic mirror descent (ASMD) (Xu et al., 2018, Alg. 2)

---

**Require:** Inputs $x^{(0)} \in \mathcal{X}, y^{(0)} = \nabla\psi(x^{(0)}) \in \mathcal{X}^*$, $A_0 = s_0 = 1/2$,
 1: **for** $k \in \mathbb{N}$ **do**
 2:    $A_{k+1} = (k+1)(k+2)/2$, $\tau_k = (A_{k+1} - A_k)/A_k$, $s_k = (k+1)^{3/2}$
 3:    $x^{(k+1)} = \frac{\tau_k}{\tau_k+1}\nabla\psi^*(y^{(k)}) + \frac{1}{\tau_k+1}x^{(k)}$
 4:    $y^{(k+1)} = y^{(k)} - \frac{A_{k+1}-A_k}{s_k}\mathsf{G}(x^{(k+1)}, \xi^{(k+1)})$
 5: **end for**

---

The corresponding optimality conditions become

$$\nabla\psi^*(y^{(k)}) = x^{(k+1)} + \frac{A_k}{A_{k+1} - A_k}(x^{(k+1)} - x^{(k)}) - U^{(k)}, \tag{18}$$

$$y^{(k+1)} - y^{(k)} = -\frac{A_{k+1} - A_k}{s_k}\mathsf{G}(x^{(k+1)}, \xi^{(k+1)}). \tag{19}$$

We can perform a similar convergence analysis using the energy function given in Xu et al. (2018). Let $\mathcal{E}^{(k)} = \mathbb{E}[A_k(f(x^{(k)}) - f(x^*)) + s_k B_\psi(x^*, \nabla\psi^*(y^{(k)}))]$. We require a bound on the diameter of the optimization domain, as stated in the following assumption.

**Assumption 3.** *There exists a constant $M_\psi > 0$ such that $M_\psi = \sup_{x,x' \in \mathcal{X}} B_\psi(x, x')$.*

**Theorem 3.** *Suppose $f$ has $L_f$-Lipschitz gradient, the mirror map is $\alpha$-strongly convex, the approximation error $U^{(k)}$ is bounded, and that the stochastic oracle is otherwise unbiased with bounded second moments. Assume Assumption 3 holds, and suppose further that there exists a constant $K$ such that for every iterate $x^{(k)}$, we have*

$$\langle \nabla f(x^{(k)}), U^{(k)} \rangle \le K.$$

*Then the convergence rate of approximate ASMD is,*

$$\mathbb{E}[f(x^{(k)}) - f(x^*)] \le K + \mathcal{E}^{(0)}/A_k + \frac{(k+1)^{3/2}}{A_k}\left[M_\psi + \frac{8L_f^2 M_\psi + 2\alpha\sigma^2 + 4\|\nabla f(x^*)\|_*^2}{\alpha^2}\right] \tag{20}$$

$$= K + \mathcal{O}(k^{-2} + k^{-1/2}).$$

*Proof.* Deferred to Appendix C. $\qquad\square$

Theorem 3 thus extends the classical $\mathcal{O}(k^{-2} + k^{-1/2})$ convergence rate for accelerated stochastic mirror descent to the approximate case, up to a constant depending on the approximation error. We note that $f$ having bounded gradients is sufficient to satisfy the second assumption that $\langle \nabla f(x^{(k)}), U^{(k)} \rangle$ is bounded, as we have also assumed that $U^{(k)}$ is bounded.

## 5 Experiments

In this section, we demonstrate the performance of the proposed algorithms on image reconstruction and various training tasks. As we will be using neural networks to model the mirror potentials, we refer to the three proposed algorithms as learned AMD (LAMD), learned SMD (LSMD), and learned ASMD (LASMD) for the algorithms proposed in Sections 2 to 4 respectively with the learned mirror potentials. We will primarily consider GD, Nesterov accelerated GD, and the Adam optimizer as baselines for these tasks, with SGD where stochasticity is applicable. A grid search was used to find the baseline optimizer step-sizes, chosen to minimize the loss at iteration 100. Specific details for the choice of baseline step-sizes can be found in Appendix E. In the image reconstruction experiments, proximal operators are easily available, which allows us to compare with a learned primal-dual (LPD) method, given by Algorithm 4.4 in Banert et al. (2020). This LPD scheme can be interpreted as a generalization of the PDHG and Douglas–Rachford algorithms.

For fairness, we use pre-trained mirror maps as given in Tan et al. (2023b;a), and replicate selected experiments from these works to review the convergence behavior as compared to LMD. The mirror maps were trained to minimize a loss function of the form

$$\tilde{L}(\theta, \vartheta) = \sum_{k=1}^{N} \left[ f(\tilde{x}_k^{(i)}) + s_k \|(\nabla M_\vartheta^* \circ \nabla M_\theta - I)(\tilde{x}_k^{(i)})\| \right], \tag{21}$$

where $\tilde{x}_k^{(i)}$ are the approximate MD iterations as in Equation (4), and the mirror maps are tied across each iteration. The loss $\tilde{L}$ is averaged over a family of training objectives $f$, such as TV-regularized image reconstruction or SVM hinge losses, and the regularization parameter $s_k$ is increased as training progresses. The maximum unrolled iterations is set to $N = 10$, and the LMD step-sizes are also learned for these iterations (Tan et al., 2023b).

All implementations were done in PyTorch, and training was done on Quadro RTX 6000 GPUs with 24GB of memory (Paszke et al., 2019).

## 5.1 Numerical Considerations and Implementation

For numerical stability, we additionally store the dual variables $\nabla M_\theta(x^{(k)})$, similarly to ASMD in Algorithm 3. This avoids computing $\nabla M_\theta \circ \nabla M_\vartheta^*$, which would be the identity in the exact MD case where $M_\vartheta^*$ is the convex conjugate of $M_\theta$, and appears to be a source of instability. For example, the current LMD scheme (with step-size $t$) computes

$$x^{(k+1)} = \nabla M_\vartheta^* \left( \nabla M_\theta(x^{(k)}) - t \nabla f(x^{(k)}) \right). \tag{22}$$

We propose to replace this with updates in the dual

$$y^{(k+1)} = y^{(k)} - t \nabla f(\nabla M_\vartheta^*(y^{(k)})), \ y^{(k)} = \nabla M_\theta(x^{(k)}). \tag{23}$$

In the case where $\nabla M_\vartheta^* = (\nabla M_\theta)^{-1}$, both schemes correspond exactly to mirror descent. The key difference is that the inconsistency is now within the $\nabla f$ which would heuristically be close to 0 around the minimum, instead of on the entire dual term. To put this in terms of forward-backward error, let $\hat{x}^{(k+1)} = \nabla M_\theta^{-1} \left( \nabla M_\theta x^{(k)} - t \nabla f(x^{(k)}) \right)$ be the exact mirror update on $x^{(k)}$ with mirror potential $M_\theta$. The forward-backward inconsistencies $\nabla M_\theta(x^{(k+1)}) - \nabla M_\theta(\hat{x}^{(k+1)})$ for Equation (22) and Equation (23) would then be given respectively as

$$\nabla M_\theta(x^{(k+1)}) - \nabla M_\theta(\hat{x}^{(k+1)}) = (\nabla M_\vartheta^* - (\nabla M_\theta)^{-1}) \left( \nabla M_\theta x^{(k)} - t \nabla f(x^{(k)}) \right) \quad \text{for the primal update,} \tag{24}$$

$$\nabla M_\theta(x^{(k+1)}) - \nabla M_\theta(\hat{x}^{(k+1)}) = t \nabla f(x^{(k)}) - t \nabla f(\nabla M_\vartheta^* \circ \nabla M_\theta x^{(k)}) \quad \text{for the dual update.} \tag{25}$$

By pulling the difference between $\nabla M_\vartheta^*$ and $(\nabla M_\theta)^{-1}$ into the dual term, we empirically observe a much lower forward-backward error, which would correspond to smaller constants and tighter convergence bounds for our presented theorems. We formalize the proposed learned AMD (LAMD) with $R(x, x') = \frac{1}{2}\|x - x'\|^2$, learned SMD (LSMD), and learned ASMD (LASMD) algorithms with this dual update modification in the following Algorithms 4 to 6.

Since the original pre-trained models only have step-sizes available up to $N = 10$, we need to choose step-sizes to continue optimizing. We consider running LAMD, LSMD and LASMD for 2000 iterations with various choices of step-size. In particular, we consider three constant step-sizes $t_k = \tilde{c}$, as well as step-size regimes of the form and $t_k = c/k$ and $t_k = c'/\sqrt{k}$ for LAMD and LSMD. These step-sizes were derived from step-sizes given from the pre-trained models, given for the image denoising, image inpainting and SVM experiments in Tan et al. (2023b;a). In particular, for the constant step-size extensions, we consider taking $\tilde{c}$ to be the mean, minimum and final learned step-size, i.e. $c = \sum_{i=1}^{10} t_i/10$, $c = \min_{1 \le i \le 10} t_i$, $c = t_{10}$ respectively. For

the reciprocal step-size extensions $t_k = c/k$, we compute $c$ by taking the average $c = \sum_{i=1}^{10} it_i$. For root-reciprocal extensions $t_k = c'/k$, we similarly take $c' = \sum_{i=1}^{10} i\sqrt{t_i}$. These extensions act as best-fit constants for the given learned step-sizes.

Recall that the conditions of convergence for AMD and SMD desire a non-increasing step-size condition. Moreover, convergence of the ergodic average for SMD requires a $\ell_2$–$\ell_1$ summability condition, which is satisfied by the $t_k = c/k$ extension. We will demonstrate the behavior of these algorithms under both constant, reciprocal and root-reciprocal step-size regimes.

We note that while ASMD does not have a step-size in its definition, we can artificially introduce a step-size by adding a step-size in the dual update step in Step 4 of Algorithm 3, as

$$y^{(k+1)} = y^{(k)} - t_k \frac{A_{k+1} - A_k}{s_k} \mathsf{G}(x^{(k+1)}, \xi^{(k+1)}).$$

In the case where $t_k = t$ is constant, this can be interpreted as instead running ASMD without this step-size modification on the scaled function $tf$. Therefore, we still get convergence guarantees, up to a constant multiple factor.

---

**Algorithm 4** Learned AMD (LAMD)

---

**Require:** Input $\tilde{x}^{(0)} = \tilde{z}^{(0)} = x^{(0)} \in \mathcal{X}$, parameter $r \geq 3$, step-sizes $t_k$, number of iterations $K$
1: $z^{(0)} = \nabla M_\theta(\tilde{z}^{(0)})$
2: **for** $k = 0, ..., K$ **do**
3:      $x^{(k+1)} = \lambda_k \nabla M_\vartheta^*(z^{(k)}) + (1 - \lambda_k)\tilde{x}^{(k)}$ with $\lambda_k = \frac{r}{r+k}$
4:      $z^{(k+1)} = z^{(k)} - \frac{kt_k}{r}\nabla f(x^{(k+1)})$
5:      $\tilde{x}^{(k+1)} = x^{(k+1)} - \gamma t_k \nabla f(x^{(k+1)})$
6: **end for**
7: **return** $x^{(K+1)} = \lambda_K \nabla M_\vartheta^*(z^{(K)}) + (1 - \lambda_K)\tilde{x}^{(K)}$

---

**Algorithm 5** Learned SMD (LSMD)

---

**Require:** Input $x^{(0)} \in \mathcal{X}$, step-sizes $t_k$, number of iterations $K$, stochastic gradient oracle $\mathsf{G}$
1: $y^{(0)} = \nabla M_\theta(x^{(0)})$
2: **for** $k = 0, ..., K$ **do**
3:      $y^{(k+1)} = y^{(k)} - t_k \mathsf{G}(\nabla M_\vartheta^*(y^{(k)}), \xi^{(k)})$
4: **end for**
5: **return** $x^{(K+1)} = \nabla M_\vartheta^*(y^{(K+1)})$

---

**Algorithm 6** Learned ASMD (LASMD)

---

**Require:** Input $x^{(0)} \in \mathcal{X}$, $A_0 = s_0 = 1/2$, step-sizes $t_k$, number of iterations $K$
1: $y^{(0)} = \nabla M_\theta(x^{(0)})$
2: **for** $k = 0, ..., K$ **do**
3:      $A_{k+1} = (k+1)(k+2)/2$, $\tau_k = (A_{k+1} - A_k)/A_k$, $s_k = (k+1)^{3/2}$
4:      $x^{(k+1)} = \frac{\tau_k}{\tau_k+1}\nabla M_\vartheta^*(y^{(k)}) + \frac{1}{\tau_k+1}x^{(k)}$
5:      $y^{(k+1)} = y^{(k)} - t_k \frac{A_{k+1} - A_k}{s_k} \mathsf{G}(x^{(k+1)}, \xi^{(k+1)})$
6: **end for**
7: **return** $x^{(K+1)} = \frac{\tau_K}{\tau_K+1}\nabla M_\vartheta^*(y^{(K)}) + \frac{1}{\tau_K+1}x^{(K)}$

---

## 5.2 Image Denoising

We consider image denoising on an ellipse data set, with images of size $128 \times 128$, where we apply LAMD, LSMD and LASMD with pre-trained mirror maps given in Tan et al. (2023a). The mirror maps $\nabla M_\theta$ :

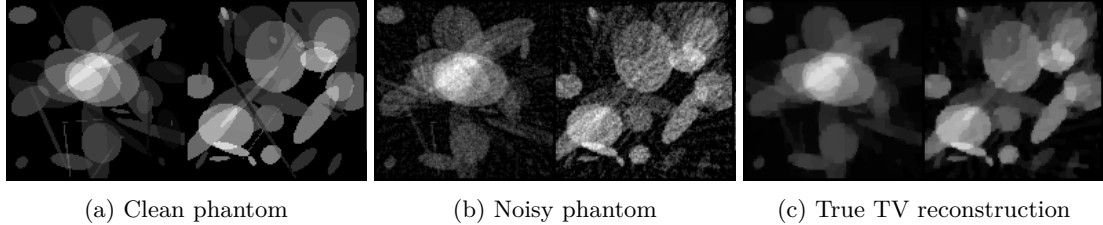

(a) Clean phantom           (b) Noisy phantom          (c) True TV reconstruction

Figure 1: Example of the ellipse phantom data, as well as the reconstruction gained by using gradient descent on the TV function.

$\mathcal{X} \to \mathcal{X}^*, \nabla M_\vartheta^* : \mathcal{X}^* \to \mathcal{X}$ are modelled using input-convex neural networks (ICNNs) (Amos et al., 2017), and are trained to minimize the function values while also minimizing the *forward-backward inconsistency*, which is defined to be $(\nabla M_\vartheta^* \circ \nabla M_\theta - I)(x^{(k)})$. The training data used for the pre-trained mirror map are noisy ellipse phantoms in X-ray CT, generated using the Deep Inversion Validation Library (DIVal) (Wang & Zhou, 2006; Leuschner et al., 2019). The phantoms were generated in the same manner as in Tan et al. (2023a). The target functions to optimize were given by TV model-based denoising,

$$\mathcal{F} = \left\{ f(x) = \|x - y\|_2^2 + \lambda \|\nabla x\|_1 \right\},$$

where $y$ are noisy phantoms, and $\nabla x$ is the pixel-wise discrete gradient, with the regularization parameter $\lambda = 0.15$.

Figure 1 presents some of the noisy ellipse data that we aim to reconstruct using a TV model. We observe that the noise artifacts are non-Gaussian and have some ray-like artifacts. We model the stochasticity by artificially adding Gaussian noise to the gradient $\mathsf{G}(x, \xi) - g(x) = \Delta(x, \xi) \sim \mathcal{N}(0, \sigma^2 I_{n \times n})$, with $\sigma = 0.05$.

Figure 2 demonstrates the effect of extending the learned step-sizes for LMD, LAMD, LSMD and LASMD, compared to the baseline optimization algorithms GD, Nesterov accelerated GD, and Adam, and LPD. We observe that step-sizes following $t_k = c/k$ perform well for LAMD and LSMD, which is consistent with observations made in Tan et al. (2023a), Theorem 2. For LASMD, we observe that a constant step-size extension choice is optimal. We also observe the poor performance of Adam, for which various non-convergence results are known for convex settings (Reddi et al., 2018; Zou et al., 2019). The plateauing phenomenon with the baselines can not be avoided in this setting, with smaller step-sizes delaying the "decrease phase" and lowering the eventual plateau. Moreover, LPD also plateaus, possibly due to non-generalization of the learned method to higher iteration counts. We will use the aforementioned optimal step-size extensions as found in Figure 2 to extend step-sizes for the following experiments.

### 5.3 Image Inpainting

We additionally consider the image inpainting setting in Tan et al. (2023b). The images to reconstruct are STL-10 images, which are $96 \times 96$ color images. The images are corrupted with a fixed 20% missing pixel mask $Z$, then 5% Gaussian noise is added to all color channels to get the noisy image. The function class we optimize over is the class of TV-regularized variational forms, with regularization parameter $\lambda = 0.15$ as chosen in Tan et al. (2023b), given by

$$\mathcal{F} = \left\{ f(x) = \|Z \circ (x - y)\|_2^2 + \lambda \|\nabla x\|_1 \right\},$$

where $y$ are images corrupted with missing pixels. We use mirror maps that are pre-trained with the training regime given in Tan et al. (2023b), and measure their performance when applied with LAMD, LSMD and LASMD as compared to MD.

We first note that this setting does not admit a stochastic interpretation, and thus we set the artificial noise in LSMD and LASMD to zero. Notice that in this case, LSMD updates revert to mirror descent updates. The main difference between the LSMD that we will present here with the LMD used in Tan et al. (2023b) is that in this work's computations, the dual iterates are saved, as described at the start of Section 5.

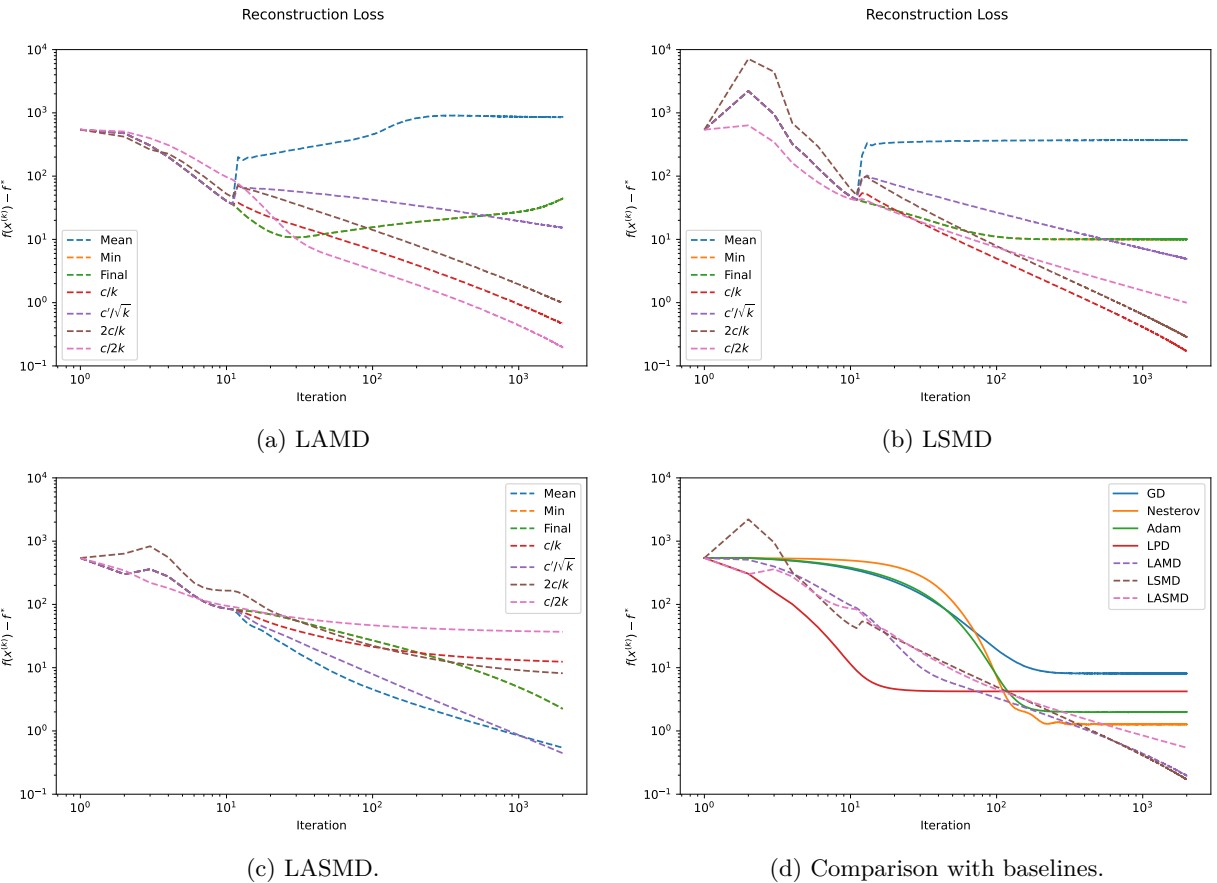

(a) LAMD

(b) LSMD

(c) LASMD.

(d) Comparison with baselines.

Figure 2: Evolution of losses for the proposed LAMD, LSMD and LASMD methods with various step-size extensions (a-c), as well as baselines, evaluated over 2000 iterations for TV model-based denoising of noisy ellipse phantoms. We observe that reciprocal step sizes generally work well for LAMD and LSMD, with larger step-sizes (given by the "mean" plot) resulting in non-convergence to minima. Constant step-sizes work better for LASMD, which can be attributed to the already decaying step-sizes in LASMD. We observe that the learned MD methods exhibit similar convergence behavior, which may be due to the relative strong convexity of the underlying optimization problem.

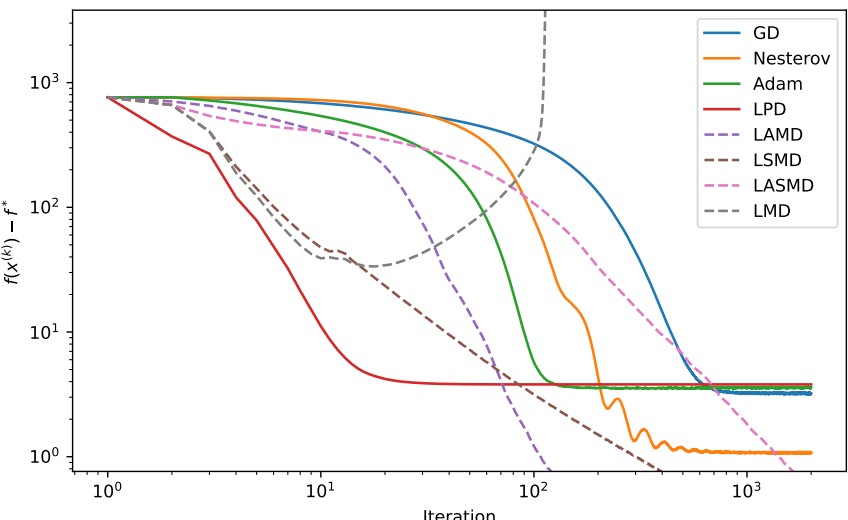

Figure 3: Plot of the function optimality gap $f(x^{(k)}) - f^*$ for 2000 iterations of various MD methods for image inpainting. The step-size regime is reciprocal for LAMD, LSMD and LMD, and constant for LASMD. We observe that each of LAMD, and LASMD both exhibit approximately $\mathcal{O}(k^{-2})$ convergnce. LMD and LSMD exhibit approximately $\mathcal{O}(k^{-1})$ convergence, with instability around iteration 100 for LMD causing the increase in function value. As we are not applying added stochastic noise in this setting, the only difference between LSMD and LMD is that the dual variables are stored as described in Section 5, which allows the LSMD iterates to stay bounded.

For comparison with baselines, we apply the proposed methods to optimize a function that arises from TV-based variational model for a single noisy masked image from the STL-10 test data set. The step-sizes were chosen by considering the best step-size extension out of those described at the start of Section 5, which were reciprocal step-size extensions for LAMD, LSMD and LMD, and constant step-size for LASMD given by the mean of the learned step-sizes. To compute the global minimum of the optimization problem, we run gradient descent for 15000 iterations with a step-size of $5 \times 10^{-4}$, followed by another 5000 iterations with a step-size of $1 \times 10^{-4}$.

Figure 3 demonstrates approximately that LAMD and LASMD are able to achieve $\mathcal{O}(k^{-2})$ convergence. Moreover, we see the effect of storing the dual iterates instead of the primal iterates, as the LSMD converges without the divergent behavior of LMD. Moreover, the convergence rate of LAMD is faster than is suggested by Theorem 1 for step-size $t_k = \Theta(k^{-1})$. This suggests that the approximation error term given in Equation (8),

$$\left\langle \tilde{z}^{(k+1)} - x^*, \nabla \psi(\tilde{z}^{(k+1)}) - \nabla \psi(\hat{z}^{(k+1)}) \right\rangle,$$

is decaying as well. This could be attributed to the iterate approaching the global minimum such that the first term $\tilde{z}^{(k+1)} - x^*$ is decaying. We additionally observe the speedup of LAMD compared to Nesterov accelerated GD and Adam in the earlier iterates, coming from the learned mirror maps, as well as better generalization performance to higher iterations as compared to LPD.

## 5.4 SVM Training

To demonstrate an application of stochasticity, we consider training a support vector machine (SVM) over a dataset with many samples. Similarly to Tan et al. (2023b), the task is to classify the digits 4 and 9 from the MNIST data set. A 5-layer neural network was used to compute feature vectors $\phi_i \in \mathbb{R}^{50}$ from the MNIST images for classification. The SVM training problem in this case is to find weights $\mathbf{w} \in \mathbb{R}^{50}$ and a bias $b$ such

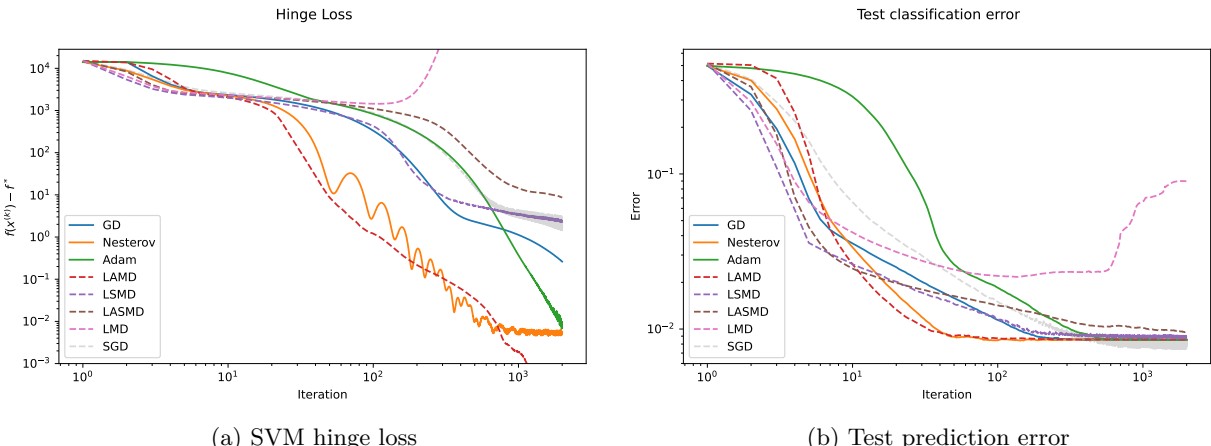

(a) SVM hinge loss

(b) Test prediction error

Figure 4: Plots of the SVM hinge loss and test accuracy for the various optimization methods tested. We observe that LAMD is able to outperform the baselines and SGD in terms of hinge loss, without the plateauing behavior of Nesterov accelerated GD. We additionally observe that the end fluctuations of LSMD are smaller due to the reciprocal step-size regime chosen. However, SGD is able to obtain solutions that have lower classification error.

that for a given set of features and targets $(\phi_i, y_i)_{i \in \mathcal{I}} \in \mathbb{R}^{50} \times \{\pm 1\}$, the prediction $\text{sign}(\mathbf{w}^\top \phi_i + b)$ matches $y_i$ for most samples. This can be reformulated into a variational problem as follows, where $C > 0$ is some positive constant,

$$\min_{\mathbf{w}, b} \frac{1}{2} \mathbf{w}^\top \mathbf{w} + \frac{C}{|\mathcal{I}|} \sum_{i \in \mathcal{I}} \max(0, 1 - y_i(\mathbf{w}^\top \phi_i + b)). \tag{26}$$

Using the pre-trained neural network, we train SVMs using the training fold of MNIST using the 4 and 9 classes, which consists of 11791 images. For testing the generalization accuracy of these methods, we use the 4 and 9 classes in the testing fold, which consists of 1991 images. For the stochastic methods, we consider using a batch-size of 500 random samples from the testing fold, and the same for stochastic gradient descent. To train the SVMs, full-batch gradient is used for LAMD, LMD, GD and Nesterov accelerated GD. Batched gradients with batch-size 500 were used for LSMD, LASMD and SGD. We consider the same number of optimization for both the full-batch methods and the stochastic variants. This is slightly different from typical learned schemes where if the data is batched, then more training iterations have to be taken to keep the number of "epochs" the same.

The optimization problem and initialization was taken to be the same as in Tan et al. (2023b), where 100 SVMs are initialized using $(\mathbf{w}, b) \sim \mathcal{N}(0, I_{50+1})$. Each SVM has its parameters optimized with respect to the hinge loss (26) using the various optimization methods. As in the previous section, we consider running each of the methods with all the step-size choices, and present the best choice for comparison. We note that the LPD scheme is not applicable in this setting as the sum in (26) does not admit an easily computable proximal term.

Figure 4 compares the losses on the entire training set, as well as the generalization error of the computed SVMs on the test set. The step-size regimes chosen are reciprocal for LAMD, LSMD and LMD, and constant for LASMD and SGD. LSMD is able to perform on par with full-batch GD despite only having access to stochastic gradients, until eventually plateauing around the same loss as SGD. We observe significant acceleration of LAMD compared to all the compared methods. LAMD also has almost identical final test classification error to the baseline methods.

# 6 Equivariant MD for NNs

In the previous sections, we demonstrated some applications of LMD for convex optimization. However, the mirror map models used still require hundreds of thousands of parameters. Compared to classical optimizers such as SGD or Adam with less than 10 hyperparameters, this is a significant limitation in scaling LMD to complex problems such as training neural networks. In this section, we propose to remedy this by reducing the number of learnable parameters for LMD while retaining expressiveness. This is done by exploiting symmetries of the problems at hand, sometimes known as *equivariance* in the literature.

Equivariance is a group-theoretic concept, where a group action commutes with a function application. In practical terms, symmetries can be exploited to reduce problem complexity. For example, translations and rotations of medical histopathology or electron microscopy images can also be considered as images from the same distribution. Bekkers et al. (2018) exploit this to construct convolutional neural networks exploiting this symmetry to achieve improved performance on various medical imaging segmentation tasks.

In the context of L2O, we aim to exploit symmetries to construct simpler or faster optimizers. Equivariance for neural networks have recently been explored, from both an external viewpoint of optimization or training losses, as well as the internal viewpoint of learned filters. Lenc & Vedaldi (2015) consider using equivariance to interpret the filters learned in a convolutional neural network. Chen et al. (2021) consider using equivariance in the unsupervised learning setting, using known equivariance properties to construct virtual operators and drastically increase the reconstruction quality. A major field is the usage of equivariant neural networks that enforce some type of group symmetry by using specific network architectures and activations, used in inverse problems and graph learning problems (Celledoni et al., 2021; Cohen & Welling, 2016; Worrall et al., 2017). By strictly enforcing symmetry, the resulting networks are more measurement-consistent and have better reconstruction quality. We focus on the use of equivariance for dimensionality reduction, provably demonstrating "weight-tying" when using LMD for neural networks.

To simplify the process of training LMD for neural networks, we consider exploiting the architecture of a neural network. In particular, the weights of a neural network typically carry permutation symmetries, which are exploited in works such as the Neural Tangent Kernel or Neural Network Gaussian Processes (Jacot et al., 2018; Lee et al., 2018). For example, a neural network with a dense layer and layer-wise activations between two feature layers will be invariant to permutations of the weights and feature indices. Another example is convolutional networks, where permuting the kernels will result in a permuted feature layer. In this section, we aim to formalize equivariance in the context of neural networks, and eventually show that we can MD methods that are equivariant stay equivariant along their evolution. This means that we can replace parameterizations of MD with equivariant parameterizations, reducing the number of learnable parameters while retaining the expressivity.

Let us first formalize the notion of symmetry that we aim for equivariance with. Let $G$ be a group of symmetries acting linearly on a real Hilbert parameter space $(\mathcal{Z}, \langle \cdot, \cdot \rangle)$. In other words, $G$ acts on $\mathcal{Z}$ via the binary operator $\cdot : G \times \mathcal{Z} \to \mathcal{Z}$, and moreover, for any $z, z' \in \mathcal{Z}$, $\lambda \in \mathbb{R}$, we have

$$g \cdot (z + z') = g \cdot z + g \cdot z', \, g \cdot (\lambda z) = \lambda g \cdot z.$$

Assume further that $G$ has an adjoint in the inner product, i.e. $\langle g \cdot z, z' \rangle = \langle z, g^{-1} \cdot z' \rangle$. One example is when $\mathcal{Z} = \mathbb{R}^n$ equipped with the standard inner product, and $G$ is a subset of the orthogonal group $O(n)$ (the group of $n \times n$ orthonormal matrices) with the natural action (matrix multiplication).

Let $L : \mathcal{Z} \to \mathbb{R}$ be a loss function for our parameters. Suppose $x^{(0)}$ is initialized with some distribution $x^{(0)} \sim (\mathcal{Z}, p)$ that is stable under $G$, i.e., $p(x^{(0)}) = p(g \cdot x^{(0)}) \, \forall g \in G$, and further that $L$ is stable under the permutations $L(g \cdot x) = L(x)$. For example, for a feature vector $x \in \mathbb{R}^n$, the loss function $L(x) = \|x\|_2$ and initialization distribution $\mathcal{N}(0, I_n)$ are invariant under the orthogonal group $G = O(n)$. The loss functions $L(x) = \|x\|_1, \|x\|_\infty$ and any component-wise i.i.d. distribution are invariant under the permutation group $G = \mathrm{Sym}(n)$.

Recall that the objective of LMD is to minimize the loss after some number of optimization iterations (Tan et al., 2023b). To this end, let $\Omega : \mathcal{Z} \to \mathcal{Z}$ be an optimization iteration. For example, gradient descent on $L$

can be written as $\Omega(z) = z - \eta \nabla L(z)$. The objective of LMD can be written, for example, as

$$\min_{\Omega} \mathbb{E}\left[\sum_{i=1}^{N} L(z_i)\right].$$

Suppose that we optimize $\Omega$ using some sort of gradient descent (in, say, the Sobolev space $H^1(\mathcal{Z})$ of $L^2$ functions with $L^2$ derivative), and further that $\Omega_0$ satisfies

$$\Omega_0(g \cdot z) = g \cdot [\Omega_0(z)], \quad \forall g \in G, z \in \mathcal{Z}.$$

This reads that $\Omega_0$ is $G$-invariant under conjugation, where the conjugation action of $G$ on functions $F : \mathcal{Z} \to \mathcal{Z}$ is $(g \cdot F)(z) = g \cdot (F(g^{-1} \cdot z))$. We equivalently denote this as $G$-equivariance, as $g$ commutes with $\Omega_0$. For a sanity check, consider the case where $\Omega_0$ is gradient descent with on the objective function $L$. The following proposition says that $\Omega_0$ is indeed $G$-equivariant under these conditions.

**Proposition 2.** *If $\Omega_0$ is gradient descent on the $G$-invariant objective function $L$, then it is $G$-equivariant, i.e. $\Omega_0(g \cdot z) = g \cdot [\Omega_0(z)]$ for all $g \in G, z \in \mathcal{Z}$.*

*Proof.* Relies on the Riesz representation theorem and adjoint property. Deferred to Appendix D.1. $\quad\square$

Suppose first that $N = 1$, so that we want to minimize (initialized at a $G$-invariant $\Omega = \Omega_0$)

$$\mathcal{L}(\Omega) := \mathbb{E}_{z_0 \sim p}\left[L(z_1)\right] = \mathbb{E}_{z_0 \sim p}\left[L(\Omega(z_0))\right]. \tag{27}$$

We wish to show that after optimization, $\Omega$ commutes with $g$ (i.e. $\Omega$ stays invariant under the conjugation action of $G$ on functions $\mathcal{Z} \to \mathcal{Z}$). To do this, we observe that the discrete differences as computed by autograd are invariant under the group action of $g$. Assume further $\Omega_0 \in L^2(\mathcal{Z}, \mathcal{Z}, p)$ where $p$ is the initialization probability measure. The corresponding inner product is

$$\langle \Omega, \Xi \rangle_{L^2(\mathcal{Z},\mathcal{Z},p)} = \mathbb{E}_{z \sim p}[\langle \Omega(z), \Xi(z) \rangle_{\mathcal{Z}}].$$

$G$ naturally acts on $L^2(\mathcal{Z}, \mathcal{Z}, p)$ by the conjugation action

$$(g \cdot \Omega)(z) = g \cdot [\Omega(g^{-1} \cdot z)].$$

To check that this conjugation is an action, first note that the action by $g$ is isotropic since $G \leq O(n)$:

$$\int \|(g \cdot \Omega)(z)\|^2 dp(z) = \int \|(g \cdot \Omega)(z)\|^2 dp(z)$$

$$= \int \|g \cdot (\Omega(g^{-1}(z)))\|^2 dp(g \cdot z)$$

$$= \int \|\Omega(z)\|^2 dp(z).$$

Here, we used invariance of $g$ under $p$, and also that $\|g \cdot y\|^2 = \langle g \cdot y, g \cdot y \rangle = \langle y, y \rangle = \|y\|^2$ from the adjoint. The required composition and identity properties of the action are clear. Moreover, the action is linear, and we have the adjoint property

$$\langle g \cdot \Omega, \Xi \rangle_{L^2(p)} = \int \langle g \cdot \Omega(g^{-1} \cdot z), \Xi(z) \rangle dp(z)$$

$$= \int \langle \Omega(g^{-1} \cdot z), g^{-1} \cdot \Xi(z) \rangle dp(z)$$

$$= \int \langle \Omega(z), g^{-1} \Xi(g \cdot z) \rangle dp(z)$$

$$= \langle \Omega, g^{-1} \cdot \Xi \rangle.$$

Assume that $\Omega_0$ is $G$-equivariant. Then the next iterate satisfies the following

$$\Omega_1 = \Omega_0 - \eta \frac{d\mathcal{L}}{d\Omega}(\Omega_0). \tag{28}$$

But we also have for any $g$ and $\Omega$,

$$\begin{aligned}
\mathcal{L}(\Omega) &= \mathbb{E}_{z_0 \sim p}[L(\Omega(z_0))] \\
&= \mathbb{E}_{z_0 \sim p}[L(g \cdot \Omega(z_0))] && \text{since } L(z) = L(g \cdot z) \\
&= \mathbb{E}_{z_0 \sim p}[L(g \cdot \Omega(g^{-1} \cdot z_0))] && \text{since } p(z) = p(g \cdot z) \\
&= \mathbb{E}_{z_0 \sim p}[L((g \cdot \Omega)(z_0))] \\
&= \mathcal{L}(g \cdot \Omega).
\end{aligned}$$

Hence if $\Omega_0 = g \cdot \Omega_0$ for all $g$, then

$$\frac{d}{d\Omega}\mathcal{L}(\Omega_0) = \frac{d}{d(g \cdot \Omega)}\mathcal{L}(\Omega_0) \tag{29}$$

Since the variation of the $\Omega$ and $g \cdot \Omega$ is the same at $\Omega_0$, we have that $g \cdot \Omega_1 = \Omega_1$, i.e. $\Omega_1$ is $G$-equivariant. This argument can be repeated to get that $\Omega_n$ is $G$-equivariant (under conjugation) for all $n \in \mathbb{N}$. This is summarized in the following proposition.

**Proposition 3.** *Let $(\mathcal{Z}, \langle \cdot, \cdot \rangle)$ be a Hilbert parameter space. Suppose the following hold:*

1. *$G$ a group acting linearly on $\mathcal{Z}$ with an adjoint;*

2. *The loss function $L : \mathcal{Z} \to \mathbb{R}$ and are stable under $G$, that is, $L(g \cdot z) = L(z)$ for any $g \in G$ and $z \in \mathcal{Z}$;*

3. *The initialization distribution $z^{(0)} \sim (\mathcal{Z}, p)$ is stable under $G$, that is, the laws $p(z^{(0)})$ and $p(g \cdot z^{(0)})$ coincide for any $g \in G$.*

*Let $\Omega_0 : \mathcal{Z} \to \mathcal{Z}$ be an optimization iteration that is $G$-equivariant, such that $\Omega_0(g \cdot z) = g \cdot \Omega_0(z)$ for all $z \in \mathcal{Z}, g \in G$. If $\Omega_0$ is evolved using a gradient step (with step-size $\eta$) to minimize*

$$\mathcal{L}(\Omega) = \mathbb{E}_{z_0 \sim p}[L(z_1)],$$

*then the evolved optimizer $\Omega_1 = \Omega_0 - \eta d\mathcal{L}/d\Omega(\Omega_0)$ is also $G$-equivariant.*

### 6.1 Application: Deep Neural Networks

For the sake of exposition with a concrete example, let us consider a two-hidden-layer dense neural network of the form

$$N(x; A_i, b_i) = A_3 \sigma_2(A_2 \sigma_1(A_1 x + b_1) + b_2) + b_3,$$

where $x \in \mathbb{R}^{d_0}$, $\sigma_i$ are coordinate-wise or otherwise permutation-invariant activation functions such as softmax, and $A_i \in \mathbb{R}^{d_i \times d_{i-1}}$, $b_i \in \mathbb{R}^{d_i}$ are the weights and biases. In this case, the parameter space will be

$$\mathcal{Z} = \mathbb{R}^d, \ d = \sum_{i=1}^{3} (d_{i-1}d_i + d_i).$$

Let us consider a least-squares regression problem with data $(x_j, y_j)$, $j = 1, ..., k$, which can be written as minimizing the loss

$$L(z) = \sum_{j=1}^{k} [N(x_j; z) - y_j]^2.$$

Suppose that the weights are initialized entry-wise i.i.d. for each $A_i, b_i$, such as in common initializations where $A_i \sim \mathcal{N}(0, \eta_i^2 I_{d_{i-1}d_i \times d_{i-1}d_i})$ or $A_i \sim \text{Uniform}(-d_{i-1}^{-1/2}, d_{i-1}^{-1/2})$. Consider the symmetry group

$G \simeq \mathrm{Sym}(d_1) \times \mathrm{Sym}(d_2)$, consisting of permutations of the intermediate feature maps with appropriate permutations on the weights and biases, i.e. of permutations $\rho_1, \rho_2$ on $[d_1], [d_2]$ respectively with the group action as

$$
\begin{aligned}
g_{\rho_1,\rho_2}(z)[A_1]_{p,q} &= (A_1)_{p,\rho_1(q)}, \\
g_{\rho_1,\rho_2}(z)[A_2]_{p,q} &= (A_2)_{\rho_1(p),\rho_2(q)}, \\
g_{\rho_1,\rho_2}(z)[A_3]_{p,q} &= (A_3)_{\rho_2(p),q}, \\
g_{\rho_1,\rho_2}(z)[b_1]_q &= (b_1)_{\rho_1(q)}, \\
g_{\rho_1,\rho_2}(z)[b_2]_q &= (b_2)_{\rho_2(q)}, \\
g_{\rho_1,\rho_2}(z)[b_3]_q &= (b_3)_q.
\end{aligned}
$$

We can check directly that this group action satisfies the requirements of Proposition 3:

- (*Linearity*) Permutation of elements is linear.

- (*Initialization Invariance*) Follows from i.i.d. property of entries of $A_i, b_i$.

- (*Loss Equivariance*) Follows from $\sigma_i$ acting coordinatewise and invariance of sums under permutation.

Consider the simple approach of scaling each component of the gradient by a learned constant, i.e. MD with a mirror map of the form $\psi = x^\top D x$ for some learned positive definite diagonal $D$. Notice that starting with $D = I$ yields gradient descent, which is equivariant under $G$. Using Proposition 3, we find that $D$ will continue to be $G$-equivariant. In other words, each of the components of $D$ must be fixed under permutation actions of the form $(\rho_1, \rho_2) \in G$.

We can explicitly demonstrate this experimentally on a small-scale neural network. In particular, we consider training a classifier using a three-layer neural network with fully connected layers on the 2D moons data set. We arbitrarily choose 50 dimensions for the middle feature layer. The neural network takes the form

$$
N(x; A_i, b_i) = \sigma_2(A_2 \sigma_1(A_1 x + b_1) + b_2),
$$

where $A_i \in \mathbb{R}^{d_i \times d_{i-1}}$, $b_i \in \mathbb{R}^{d_i}$, $d_0 = 2, d_1 = 50, d_2 = 1$. In this case, we choose $\sigma_1$ to be a ReLU activation and $\sigma_2$ to be a log-softmax. We train using the binary cross-entropy loss. Note that in the case of a log-softmax, which takes the form

$$
\mathrm{LogSoftmax(x)_i} = \log\left( \frac{\exp(x_i)}{\sum_j \exp(x_j)} \right),
$$

the activation is still equivariant under permutations in $G$, as the sum of the exponentials stays the same. Therefore, loss equivariance holds and the above theory given by Proposition 3 applies.

Using the diagonal quadratic LMD introduced above, we thus expect from the equivariance theory that the diagonal weights in $D$ will be invariant under the effect of any permutations $(\rho_1)$ on $[d_1]$. Using superscripts to denote the weights corresponding to each learnable neural network component, we thus have that for any indices $i \in [d_1] = [2], j, j' \in [d_2] = [50], k \in [d_3] = [1]$:

$$
\begin{aligned}
(D^{A_1})_{i,j} &= (D^{A_1})_{i,j'} \\
(D^{b_1})_j &= (D^{b_1})_{j'} \\
(D^{A_2})_{j,k} &= (D^{A_2})_{j',k}.
\end{aligned}
$$

Therefore, from $G$-equivariance of LMD with diagonal weighting $D$, we have that the weights are constant across the orbits of $\rho_1$, and are thus fully determined by the weights

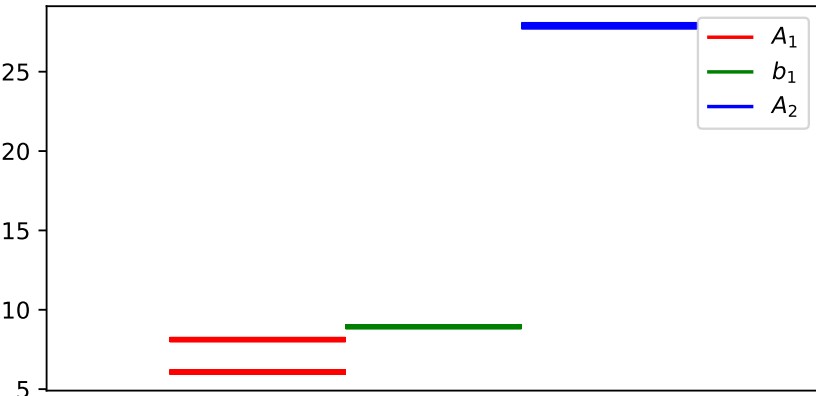

Figure 5: Componentwise gradient weightings learned for a two layer neural network on 2D moons data. Each band consists of 50 datapoints, corresponding to elements of the weight matrix $D$ for each neural network parameter group. We observe strong grouping of the weightings within each parameter group. In particular, we find that the weightings of $A_1$ are split into two groups, which correspond to the two input dimensions.

$(D^{A_1})_{1,1}, (D^{A_1})_{2,1}, (D^{b_1})_1, (D^{A_2})_{1,1}, (D^{b_2})_1$. This reduces the number of variables that need to be learned from $\dim(D) = 201$ to only 5. The equivariance effect can be clearly seen in Figure 5, which empirically demonstrates what happens if we train all the variables of $D$ without directly enforcing equivariance. By initializing $D$ to be the identity, the initial mapping $\Omega_0 : \mathcal{Z} \to \mathcal{Z}$ induced by $D$ is gradient descent, which we have shown before to be equivariant under permutations of the intermediate feature layer. The theory presented thus suggests that the optimization path of $\Omega$ (or in this case, of $D$), will continue to be equivariant under such permutations. Indeed, at the end of the optimization for $D$, we observe that the weights do indeed satisfy approximate equivariance, as demonstrated by the tight grouping phenomenon of the weights. We observe that the weights $D^{A_1}$ become two groups, and $D^{b_1}$, $D^{A_2}$ becoming one group each, which supports the theory that if the initial map $\Omega_0$ is equivariant, then so are subsequent maps after training.

We note that this analysis can be used to partially explain the existing heuristic method of applying mirror maps to neural networks, such as using $p$-norms $\psi(w) = \|w\|_p^p$ (Azizan et al., 2021), squared $p$-norms $\psi(w) = \|w\|_p^2$ (D'Orazio et al., 2021; Gunasekar et al., 2018), or element-wise maps induced by projections such as hyperbolic tangent or softmax function (Ajanthan et al., 2021). For these maps, when considering the weights as vectors, permuting components does not change the value of the mirror map (though general orthogonal transformations will). This permutation symmetry over all parameters can be thought of as a special case of the presented theory which suggests only equivariance within a layer.

## 7 Experiments: Neural Network Training using Equivariance

We consider the problem of training LMD on neural networks, based on the group equivariance demonstrated in Section 6. This is a challenging non-convex optimization task, where the approximate theory for LMD (and its extensions) does not apply, due to significant errors that may occur from small perturbations in the parameters. This motivates us to consider simpler yet exact mirror maps to learn, which can be done layer-wise by the analysis in Section 6. We demonstrate competitive performance with Adam, one of the most widely used optimizers for neural network training (Kingma & Ba, 2015).

## 7.1 Deep Fully Connected Neural Network

We first consider training a classifier neural network on the MNIST image dataset. Neural networks are usually trained using minibatched gradients, as computing gradients over the whole dataset is infeasible. This is a prime use case for the stochastic variants of LMD. The neural network to train as well as the LSMD parameterization are detailed as follows.

---

**Algorithm 7** Learned AMD with minibatched gradients (LAMD*)

---

**Require:** Input $\tilde{x}^{(0)} = \tilde{z}^{(0)} = x^{(0)} \in \mathcal{X}$, parameter $r \geq 3$, step-sizes $t_k$, number of iterations $K$
1: $z^{(0)} = \nabla M_\theta(\tilde{z}^{(0)})$
2: **for** $k = 0, ..., K$ **do**
3:      $x^{(k+1)} = \lambda_k \nabla M_\vartheta^*(z^{(k)}) + (1 - \lambda_k)\tilde{x}^{(k)}$ with $\lambda_k = \frac{r}{r+k}$
4:      $z^{(k+1)} = z^{(k)} - \frac{kt_k}{r}\nabla f_k(x^{(k+1)})$
5:      $\tilde{x}^{(k+1)} = x^{(k+1)} - \gamma t_k \nabla f_k(x^{(k+1)})$
6: **end for**
7: **return** $x^{(K+1)} = \lambda_K \nabla M_\vartheta^*(z^{(K)}) + (1 - \lambda_K)\tilde{x}^{(K)}$

---

**Neural network architecture.** The number of features in each layer of the neural network is 784–50–40–30–20–10, with dense linear layers with bias between the layers and ReLU activations, for a total of 43350 parameters. The MNIST images are first flattened (giving $28 \times 28 = 784$ features), before being fed into the network. The function objective corresponding to a neural network is the cross-entropy loss for classification over the MNIST dataset. For LSMD, we associate each layer with two splines – one for the weight matrix and one for the bias vector. This results in a total of 10 learned splines, a total of 400 learnable parameters for LSMD.

**LMD parameterization.** Utilizing the reduced parameterization as suggested by Section 6, we can consider only layer-wise mirror maps $\Psi$. This allows us to use more complicated mirror maps from $\mathbb{R} \rightarrow \mathbb{R}$. One parameterization is to use *monotonic splines* $\sigma$ as mirror maps, which are derivatives of convex mirror potentials $\nabla \psi = \sigma$. We directly model the mirror map using a knot-based piecewise-linear spline parameterization as Goujon et al. (2022), where knots are defined at pre-determined locations, and the spline is (uniquely) determined by the value of $\sigma$ at each knot. We note that since the monotonic spline $\sigma : \mathbb{R} \rightarrow \mathbb{R}$ is continuous, the corresponding mirror potential given by the integral $\psi(x) = \int_0^x \sigma(t)\,dt$ is $\mathcal{C}^1$. For these experiments, we use 41 knots equispaced on the interval $[-1, 1]$, with gap 0.05, which can be parameterized using 40 parameters. More details on the construction of $\sigma$ can be found in Appendix G.

The goal of LSMD is to train neural networks with architecture faster, given MNIST data. The function class that we wish to optimize over is thus

$$\mathcal{F} = \left\{ \sum_{k=1}^{N} \mathcal{L}(\mathcal{N}_k) \,\middle|\, \text{neural networks } \mathcal{N}_0 \right\}, \tag{30}$$

where $\mathcal{L}$ is the cross-entropy of a neural network (on an image mini-batch), and $\mathcal{N}_k$ is the neural network after evolving the parameters through LSMD for $k$ iterations. Note that the number of iterations that we can unroll is significantly higher than in the image processing examples, due to the dimensionality reduction from the equivariant theory. The minibatching process means that we are training using SMD instead of the MD framework, similarly to how SGD is used instead of GD to train neural networks on large datasets. We are allowed to use three epochs of MNIST to train, which we find to be sufficient for these experiments. The MNIST batch-size is taken to be 500, resulting in unrolling $N = 360$ steps. The neural networks $\mathcal{N}_0$ are initialized using the default PyTorch initialization, where a dense layer from $C_{\text{in}}$–$C_{\text{out}}$ features has the entries of its weight matrix $A$ and bias matrix $b$ initialized from $\text{Uniform}(-C_{\text{in}}^{-1}, C_{\text{in}}^{-1})$. We use 20 such neural network initializations as a batch for each LMD training iteration, and average the cross-entropy over each neural network as our loss function.

Figure 6 shows the performance of LSMD on this task, as well as minibatched LAMD* Algorithm 7 applied with the spline-based mirror maps learned using the LSMD framework. LMD and LAMD* are compared

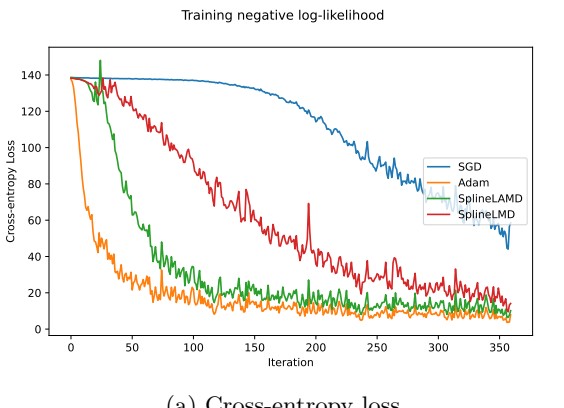
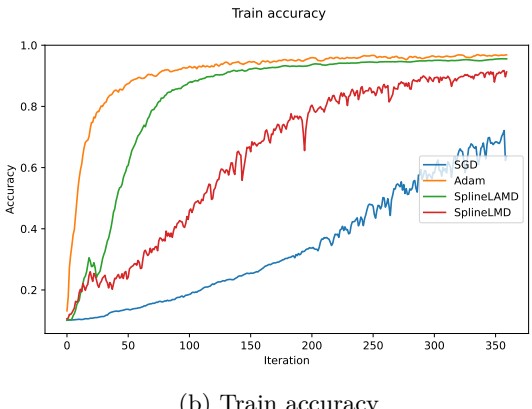

(a) Cross-entropy loss          (b) Train accuracy

Figure 6: Spline-based LMD applied to training a 4-hidden-layer dense neural network for MNIST classification. We observe a significant acceleration of LAMD as compared to LMD. There is a slight performance gap between LAMD and Adam on this task.

against: SGD with learning rate $1 \times 10^{-1}$, and Adam with learning rate $1 \times 10^{-2}$, where other parameters are kept as default. The values are averaged over 20 neural network initializations. We observe that Adam with this learning rate is able to train this network architecture very rapidly. LAMD* is able to almost match the performance of Adam at the end of the 3 training epochs, while both LAMD* and LMD significantly outperform SGD.

## 7.2 Convolutional Neural Network

We additionally consider LMD to train a 2-layer convolutional neural network. We impose a diagonal quadratic LMD prior on the weights, similarly to Section 6.1. The CNN is taken to have a single convolutional layer with 8 channels and kernel size 3, a max pooling layer with window size 2, followed by a dense layer from 288 features to 10 output features. The inputs of the CNN are MNIST images that have been downscaled to have dimension $14 \times 14$. The target is to minimize the cross-entropy loss of the CNN after 2 epochs of MNIST, with mini-batch size 1000, resulting in a total unrolling count of $N = 120$.

The equivariance here is due to the permutation invariance of each of the eight $3 \times 3$ kernels. Therefore, we can reduce the number of parameters of LMD by tying the diagonal weights across the channels, reducing the number of parameters for this layer from $8 \times (9 + 1) = 72$ to only 10, with 9 from the sliding window weights and 1 from the bias term. While not strictly induced by the training problem[1], we additionally impose equivariance on the dense layer by tying the 288 features together, reducing the number of parameters from $288 \times 10 + 10$ to $1 \times 10 + 10$. In total, we learn 30 parameters.

Figure 7 shows the performance of LMD and LAMD* when averaged over training 20 randomly initialized CNN problems, trained for 2 epochs of MNIST. We observe competitive performance with Adam for both LAMD* and LMD. Interestingly, we observe that the testing accuracy of the training result of LAMD* is marginally higher than that of Adam. This distinctly highlights the potential of learning the geometry for gradient-based methods, as we achieve significant acceleration with only 30 parameters to the point of being competitive with Adam, while retaining a principled approach to learning the model.

We note that the size of the CNN is limited due to the memory constraints of LMD. To train one iteration of LMD, all the intermediate activations of the neural network need to be stored. In the case of a convolutional layer, the intermediate layer is also image-like, which results in very high memory usage.

## 7.3 Few-Shot Learning

---

[1]Since the layer before the dense layer is a convolutional layer (followed by max-pooling), the features in this intermediate layer still contain local information. Therefore, we do not have permutation invariance of the features.

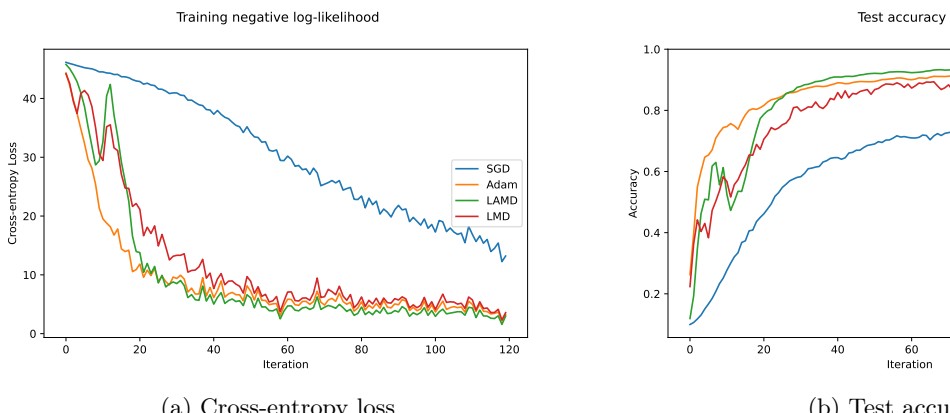

(a) Cross-entropy loss

(b) Test accuracy

Figure 7: Diagonal LMD and LAMD applied to training one-hidden-layer convolutional neural networks with 2 epochs of MNIST, averaged over 20 runs. We observe that LAMD can achieve higher test accuracy and lower training loss by the end of the 2 training epochs. Both LMD and LAMD are competitive with Adam in this example.

To further compare the proposed LMD methods on non-convex problems, we consider the problem of few-shot learning, where a classifier is trained on limited training data. We utilize the Omniglot dataset, consisting of 20 instances each of 1623 characters (Lake et al., 2011). The characters are split into training and testing sets of 1200 and 423 characters respectively as in Finn et al. (2017). We consider a 5-shot 10-way problem to utilize the small CNN architecture as detailed in the previous section. In other words, a problem training instance consists of choosing 10 characters from the respective training or testing set, training the network to classify 5 instances of each character, and testing on the other 15 instances of the character.

As a baseline meta-learning approach, we consider the Reptile method (Nichol et al., 2018), which finds a good neural network initialization such that a standard optimization algorithm such as SGD can minimize the loss quickly. Further details on how the Reptile baseline is trained can be found in Appendix H.

The L2O approach of LMD instead considers a different approach of fast optimization by changing the optimization trajectory, allowing for random neural network initializations. We consider training LMD on the training character set in two settings: one with standard random initialization, and one from the initialization learned using Reptile. These methods are then compared with SGD with learning rate $1 \times 10^{-1}$ and Adam $1 \times 10^{-2}$ as before, with random initializations and Reptile initializations. We additionally consider LAMD with the same learned parameters from both random initialization and from the Reptile initialization. The methods are evaluated on $10^4$ randomly sampled 5-shot 10-way problems from characters drawn from the test split, with gradients calculated in a full-batch manner.

Figure 8 contains a comparison of the equivariant LMD and LAMD methods with SGD and Adam, with random intializations and from the Reptile initialization. We note that LMD outperforms its SGD counterpart in both scenarios. Moreover, LAMD is able to achieve a significantly lower log-likelihood, outperforming Adam with random network initialization as well as with Reptile initializations. Nonetheless, there is still a gap between the LMD methods and Adam in terms of generalization performance, suggesting that the trajectory than LMD takes trades off generalization performance for faster training. Interestingly, LAMD with random initializations outperforms LAMD with the Reptile initialization, suggesting that random initializations plays a role in learning the geometry over a wider set of trajectories.

## 8 Conclusions

In this work, we present multiple extensions of the learned mirror descent method introduced in a recent work on learning to optimize (Tan et al., 2023b;a). In particular, we introduce accelerated and stochastic versions, as well as a new training paradigm that reduces the number of learnable parameters while maintaining

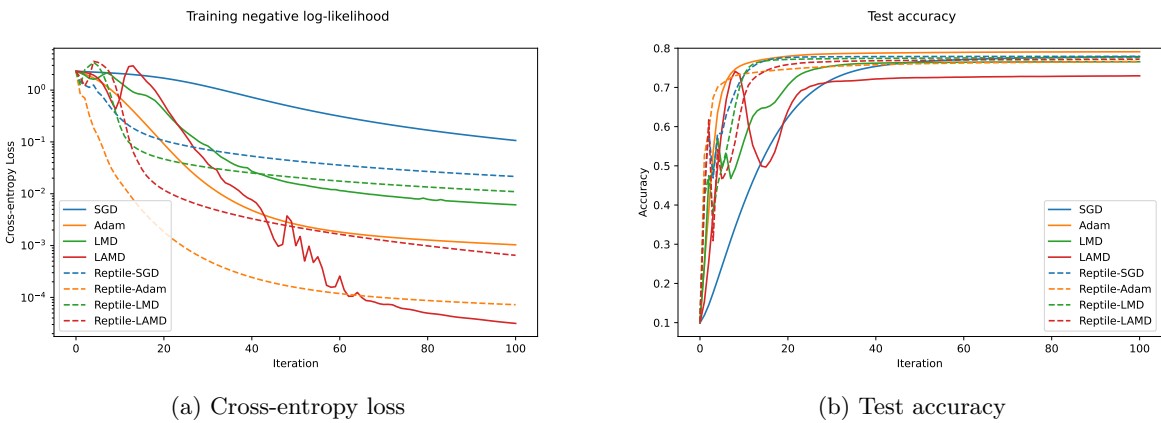

(a) Cross-entropy loss            (b) Test accuracy

Figure 8: Training loss and test accuracy for 5-shot 10-way learning on the Omniglot dataset. We observe that LMD with Reptile initialization is able to achieve similar test accuracy to the compared methods. Moreover, LAMD is able to outperform Adam in terms of training loss, but seems to converge to a solution that generalizes worse.

the *expressivity* of the mirror maps by exploiting symmetries. Numerical experiments demonstrate that the learned accelerated and stochastic MD methods are able to outperform their corresponding GD counterparts as well as previous LMD with the same mirror maps, and we empirically improve on the constant optimization gap by utilizing the dual domain during mirror steps. These experiments additionally demonstrate that mirror maps trained using LMD as in (Tan et al., 2023b;a) are able to successfully generalize to different LMD-type optimization schemes. We empirically show that mirror maps respect group invariances under training to support our equivariant theory, and exploit this to train LMD on the non-convex problems of training deep and convolutional neural networks.

This work presented results as well as techniques for developing L2O schemes for large-scale applications. Given the various modifications to LMD presented, the mirror maps were pretrained using a basic scheme that penalizes the objective function as well as the forward-backward inconsistency for a fixed number of unrolled iterates (Tan et al., 2023b). One consideration is the modification of the training scheme to instead use the accelerated or stochastic mirror descent rather than full-batch mirror descent, which could help in large-scale applications. Another possible direction would be to consider convergence properties of the approximate MD methods for non-convex functions, similarly to the non-mirrored cases presented in Berahas et al. (2017); Jin et al. (2017).

This work additionally presented an application of equivariance for dimensionality reduction of LMD, reducing the total number of parameters required to train. We demonstrated that the reduced parameterization arises naturally from common training scenarios, and that the resulting LMD is able to perform competitively with Adam. While we limit our equivariance to simple gradient-based methods, interesting future directions could consider equivariance under the scope of other optimization algorithms utilizing techniques such as scaling, momentum, or using proximal operators. Another direction to continue pushing LMD is to address the current limitations caused by needing to store intermediate activations, allowing for this method to be used for larger and more practical neural networks.

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

# A  Proofs in Section 2

## A.1  Proof of Lemma 1

**Lemma 2.** *Consider the approximate AMD iterations from Algorithm 1, and let $\hat{z}^{(k)}$ be the exact MD iterates given by Equation 5. Assume $\psi^*$ is $L_{\psi^*}$-smooth with respect to a reference norm $\|\cdot\|_*$ on the dual space, i.e. $B_{\psi^*}(z,y) \leq \frac{L_{\psi^*}}{2}\|z-y\|_*^2$, or equivalently that $\nabla\psi^*$ is $L_{\psi^*}$-Lipschitz. Assume further that there exists $0 < \ell_R \leq L_R$ such that for all $x, x' \in \mathcal{X}$, $\frac{\ell_R}{2}\|x-x'\|^2 \leq R(x,x') \leq \frac{L_R}{2}\|x-x'\|^2$. Define the energy $\tilde{E}^{(k)}$ for $k \geq 0$ as follows (where $t_{-1} = 0$):*

$$\tilde{E}^{(k)} := \frac{k^2 t_{k-1}}{r}\left(f(\tilde{x}^{(k)}) - f^*\right) + rB_{\psi^*}\left(\nabla\psi(\tilde{z}^{(k)}), \nabla\psi(x^*)\right). \tag{31}$$

*Assume the step-size conditions $\gamma \geq L_R L_{\psi^*}$, and $t_k \leq \frac{l_R}{L_f \gamma}$. Then the difference between consecutive energies satisfies the following:*

$$\tilde{E}^{(k+1)} - \tilde{E}^{(k)} \leq \frac{(2k+1-rk)t_k}{r}\left(f(\tilde{x}^{(k+1)}) - f^*\right) - \frac{k^2(t_{k-1}-t_k)}{r}\left(f(\tilde{x}^{(k)}) - f^*\right)$$
$$+ r\left\langle\nabla\psi(\tilde{z}^{(k+1)}) - \nabla\psi(\hat{z}^{(k+1)}), \tilde{z}^{(k+1)} - x^*\right\rangle.$$

*Proof.* The proof closely follows that of Lemma 2 in Krichene et al. (2015). We will use the following lemmas, which will be stated without proof. The proofs of these results can be found in the supplementary material of Krichene et al. (2015).

**Lemma 3.** *Let $f$ be convex with $L_f$-Lipschitz gradient with respect to $\|\cdot\|$. Then for all $x, x', x^+$,*

$$f(x^+) \le f(x') + \langle \nabla f(x), x^+ - x' \rangle + \frac{L_f}{2}\|x^+ - x\|^2.$$

**Lemma 4.** *For any differentiable convex $\psi^*$ and $u, v, w \in \operatorname{dom}\psi^*$,*

$$B_{\psi^*}(u, v) - B_{\psi^*}(w, v) = -B_{\psi^*}(w, u) + \langle \nabla\psi^*(u) - \nabla\psi^*(v), u - w \rangle.$$

**Lemma 5.** *If $\psi^*$ has $L_{\psi^*}$-Lipschitz gradient, then for all $u, v \in (\mathbb{R}^n)^*$,*

$$\frac{1}{2L_{\psi^*}}\|\nabla\psi^*(u) - \nabla\psi^*(v)\|^2 \le B_{\psi^*}(u, v) \le \frac{L_{\psi^*}}{2}\|u - v\|_*^2.$$

We begin our analysis with these lemmas in place. Let $x^* = z^*$ be some minimizer. We begin by bounding the difference in Bregman divergence.

$$B_{\psi^*}\left(\nabla\psi(\tilde{z}^{(k+1)}), \nabla\psi(x^*)\right) - B_{\psi^*}\left(\nabla\psi(\tilde{z}^{(k)}), \nabla\psi(x^*)\right)$$

$$\overset{(a)}{=} -B_{\psi^*}\left(\nabla\psi(\tilde{z}^{(k)}), \nabla\psi(\tilde{z}^{(k+1)})\right) + \left\langle \nabla\psi(\tilde{z}^{(k+1)}) - \nabla\psi(\tilde{z}^{(k)}), \tilde{z}^{(k+1)} - x^* \right\rangle$$

$$\overset{(b)}{\le} -\frac{1}{2L_{\psi^*}}\|\tilde{z}^{(k+1)} - \tilde{z}^{(k)}\|^2 + \left\langle \nabla\psi(\tilde{z}^{(k+1)}) - \nabla\psi(\tilde{z}^{(k)}), \tilde{z}^{(k+1)} - x^* \right\rangle$$

$$\overset{(c)}{\le} -\frac{1}{2L_{\psi^*}}\|\tilde{z}^{(k+1)} - \tilde{z}^{(k)}\|^2 + \left\langle -\frac{kt_k}{r}\nabla f(x^{(k+1)}), \tilde{z}^{(k+1)} - x^* \right\rangle + \left\langle \nabla\psi(\tilde{z}^{(k+1)}) - \nabla\psi(\hat{z}^{(k+1)}), \tilde{z}^{(k+1)} - x^* \right\rangle.$$

$$(32)$$

In Step (a) above, we used $\nabla\psi^* \circ \nabla\psi = I$ and $z^* = x^*$ together with Lemma 4. In Step (b), we use Lemma 5 to bound the first term. In Step (c), we decomposed the right-hand side of the inner product using

$$\nabla\psi(\tilde{z}^{(k+1)}) - \nabla\psi(\hat{z}^{(k+1)}) + \nabla\psi(\hat{z}^{(k+1)}) - \nabla\psi(\tilde{z}^{(k)}) = \nabla\psi(\tilde{z}^{(k+1)}) - \nabla\psi(\hat{z}^{(k+1)}) - \frac{kt_k}{r}\nabla f(x^{(k+1)}),$$

which follows directly from Equation 5. Recall Step 4 of Algorithm 1:

$$\tilde{x}^{(k+1)} = \arg\min_{\tilde{x} \in \mathbb{R}^n} \left\langle \nabla f(x^{(k+1)}), \tilde{x} \right\rangle + \frac{1}{\gamma t_k}R(\tilde{x}, x^{(k+1)}),$$

where $\frac{l_R}{2}\|x - y\|^2 \le R(x, y) \le \frac{L_R}{2}\|x - y\|^2$. From the definition of $\tilde{x}^{(k+1)}$, we have that for any $x \in \mathbb{R}^n$,

$$R(x, x^{(k+1)}) \ge R(\tilde{x}^{(k+1)}, x^{(k+1)}) + \gamma t_k\langle \nabla f(x^{(k+1)}), \tilde{x}^{(k+1)} - x \rangle. \tag{33}$$

Recalling the step for $x^{(k+1)}$ in Step 2 of Algorithm 1, we can write

$$\tilde{z}^{(k+1)} - \tilde{z}^{(k)} = \frac{1}{\lambda_k}\left(\lambda_k\tilde{z}^{(k+1)} + (1 - \lambda_k)\tilde{x}^{(k)} - x^{(k+1)}\right) = \frac{1}{\lambda_k}\left(d^{(k+1)} - x^{(k+1)}\right),$$

where we define $d^{(k+1)} := \lambda_k\tilde{z}^{(k+1)} + (1 - \lambda_k)\tilde{x}^{(k)}$. We then compute

$$\|\tilde{z}^{(k+1)} - \tilde{z}^{(k)}\|^2 = \frac{1}{\lambda_k^2}\|d^{(k+1)} - x^{(k+1)}\|^2 \ge \frac{2}{L_R\lambda_k^2}R(d^{(k+1)}, x^{(k+1)}), \text{ since } R(x, x') \le \frac{L_R}{2}\|x - x'\|^2.$$

$$\implies \|\tilde{z}^{(k+1)} - \tilde{z}^{(k)}\|^2 \ge \frac{2}{L_R\lambda_k^2}\left(R(\tilde{x}^{(k+1)}, x^{(k+1)}) + \gamma t_k\left\langle \nabla f(x^{(k+1)}), \tilde{x}^{(k+1)} - d^{(k+1)} \right\rangle\right),$$

by setting $x = d^{(k+1)}$ in Equation 33. Now, using $R(\tilde{x}^{(k+1)}, x^{(k+1)}) \ge \frac{l_R}{2}\|\tilde{x}^{(k+1)} - x^{(k+1)}\|^2$, we have:

$$\|\tilde{z}^{(k+1)} - \tilde{z}^{(k)}\|^2 \ge \frac{2}{L_R\lambda_k^2}\left(\frac{l_R}{2}\|\tilde{x}^{(k+1)} - x^{(k+1)}\|^2 + \gamma t_k\left\langle \nabla f(x^{(k+1)}), \tilde{x}^{(k+1)} - d^{(k+1)} \right\rangle\right).$$

By replacing $d^{(k+1)} = \lambda_k \tilde{z}^{(k+1)} + (1 - \lambda_k)\tilde{x}^{(k)}$ and multiplying both sides by $\lambda_k \frac{kL_R}{2r\gamma}$, we get

$$\lambda_k \frac{kL_R}{2r\gamma}\|\tilde{z}^{(k+1)} - \tilde{z}^{(k)}\|^2 \geq \frac{kl_R}{2r\lambda_k\gamma}\|\tilde{x}^{(k+1)} - x^{(k+1)}\|^2$$
$$+ \left\langle \frac{kt_k}{r}\nabla f(x^{(k+1)}), \frac{1}{\lambda_k}\tilde{x}^{(k+1)} - \tilde{z}^{(k+1)} - \frac{1-\lambda_k}{\lambda_k}\tilde{x}^{(k)} \right\rangle. \quad (34)$$

Subtracting Equation 34 from Equation 32:

$$B_{\psi^*}(\nabla\psi(\tilde{z}^{(k+1)}), \nabla\psi(x^*)) - B_{\psi^*}(\nabla\psi(\tilde{z}^{(k)}), \nabla\psi(x^*)) \leq -\alpha_k\|\tilde{z}^{(k+1)} - \tilde{z}^{(k)}\|^2 - \frac{kl_R}{2r\lambda_k\gamma}\|\tilde{x}^{(k+1)} - x^{(k+1)}\|^2$$
$$+ \left\langle \frac{-kt_k}{r}\nabla f(x^{(k+1)}), -x^* + \frac{1}{\lambda_k}\tilde{x}^{(k+1)} - \frac{1-\lambda_k}{\lambda_k}\tilde{x}^{(k)} \right\rangle + \left\langle \nabla\psi(\tilde{z}^{(k+1)}) - \nabla\psi(\hat{z}^{(k+1)}), \tilde{z}^{(k+1)} - x^* \right\rangle,$$

where $\alpha_k = \frac{1}{2L_{\psi^*}} - \frac{k\lambda_k L_R}{2r\gamma}$. Defining $D_1^{(k+1)} = \|\tilde{x}^{(k+1)} - x^{(k+1)}\|^2$ and $D_2^{(k+1)} = \|\tilde{z}^{(k+1)} - \tilde{z}^{(k)}\|^2$, we rewrite the last inequality as

$$B_{\psi^*}(\nabla\psi(\tilde{z}^{(k+1)}), \nabla\psi(x^*)) - B_{\psi^*}(\nabla\psi(\tilde{z}^{(k)}), \nabla\psi(x^*))$$
$$\leq -\alpha_k D_2^{(k+1)} - \frac{kl_R}{2r\lambda_k\gamma}D_1^{(k+1)} + \frac{kt_k}{r}\left\langle -\nabla f(x^{(k+1)}), \tilde{x}^{(k+1)} - x^* \right\rangle$$
$$+ \frac{1-\lambda_k}{\lambda_k}\frac{kt_k}{r}\left\langle -\nabla f(x^{(k+1)}), \tilde{x}^{(k+1)} - \tilde{x}^{(k)} \right\rangle + \left\langle \nabla\psi(\tilde{z}^{(k+1)}) - \nabla\psi(\hat{z}^{(k+1)}), \tilde{z}^{(k+1)} - x^* \right\rangle.$$

Using Lemma 3 with $x, x^+, x'$ as follows, we bound the inner products:

$$\left\langle -\nabla f(x^{(k+1)}), \tilde{x}^{(k+1)} - \tilde{x}^{(k)} \right\rangle \leq f(\tilde{x}^{(k)}) - f(\tilde{x}^{(k+1)}) + \frac{L_f}{2}D_1^{(k+1)}, \; (x = x^{(k+1)}, \; x^+ = \tilde{x}^{(k+1)}, \text{ and } x' = \tilde{x}^{(k)});$$
$$\left\langle -\nabla f(x^{(k+1)}), \tilde{x}^{(k+1)} - x^* \right\rangle \leq f^* - f(\tilde{x}^{(k+1)}) + \frac{L_f}{2}D_1^{(k+1)}, \; (x = x^{(k+1)}, \; x^+ = \tilde{x}^{(k+1)}, \text{ and } x' = x^*).$$

Combining the inequalities and using $\frac{1-\lambda_k}{\lambda_k} = \frac{k}{r}$, we obtain

$$B_{\psi^*}(\nabla\psi(\tilde{z}^{(k+1)}), \nabla\psi(x^*)) - B_{\psi^*}(\nabla\psi(\tilde{z}^{(k)}), \nabla\psi(x^*))$$
$$\leq -\alpha_k D_2^{(k+1)} + \frac{k^2 t_k}{r^2}\left( f(\tilde{x}^{(k)}) - f(\tilde{x}^{(k+1)}) + \frac{L_f}{2}D_1^{(k+1)} \right) + \frac{kt_k}{r}\left( f^* - f(\tilde{x}^{(k+1)}) + \frac{L_f}{2}D_1^{(k+1)} \right)$$
$$- \frac{kl_R}{2r\lambda_k\gamma}D_1^{(k+1)} + \left\langle \nabla\psi(\tilde{z}^{(k+1)}) - \nabla\psi(\hat{z}^{(k+1)}), \tilde{z}^{(k+1)} - x^* \right\rangle$$
$$= \frac{k^2 t_k}{r^2}\left( f(\tilde{x}^{(k)}) - f(\tilde{x}^{(k+1)}) \right) + \frac{kt_k}{r}\left( f^* - f(\tilde{x}^{(k+1)}) \right) - \alpha_k D_2^{(k+1)} - \beta_k D_1^{(k+1)}$$
$$+ \left\langle \nabla\psi(\tilde{z}^{(k+1)}) - \nabla\psi(\hat{z}^{(k+1)}), \tilde{z}^{(k+1)} - x^* \right\rangle,$$

where $\beta_k := \frac{kl_R}{2r\lambda_k\gamma} - \frac{L_f k^2 t_k}{2r^2} - \frac{L_f kt_k}{2r}$. Re-introducing the energy function for $k \geq 1$,

$$\tilde{E}^{(k)} := \frac{k^2 t_{k-1}}{r}(f(\tilde{x}^{(k)}) - f^*) + rB_{\psi^*}(\nabla\psi(z^{(k)}), \nabla\psi(x^*)),$$

we can compute the difference $\tilde{E}^{(k+1)} - \tilde{E}^{(k)}$:

$$
\begin{aligned}
\tilde{E}^{(k+1)} - \tilde{E}^{(k)} &= \frac{(k+1)^2 t_k}{r} \left( f(\tilde{x}^{(k+1)}) - f^* \right) - \frac{k^2 t_{k-1}}{r} \left( f(\tilde{x}^{(k)}) - f^* \right) \\
&\quad + r(B_{\psi^*}(\nabla\psi(\tilde{z}^{(k+1)}), \nabla\psi(x^*)) - B_{\psi^*}(\nabla\psi(\tilde{z}^{(k)}), \nabla\psi(x^*))) \\
&= \frac{k^2 t_k}{r} \left( f(\tilde{x}^{(k+1)}) - f^* \right) + \frac{(2k+1)t_k}{r} \left( f(\tilde{x}^{(k+1)}) - f^* \right) \\
&\quad - \frac{k^2 t_k}{r} \left( f(\tilde{x}^{(k)}) - f^* \right) - \frac{k^2 t_{k-1} - k^2 t_k}{r} \left( f(\tilde{x}^{(k)}) - f^* \right) \\
&\quad + \frac{k^2 t_k}{r} \left( f(\tilde{x}^{(k)}) - f(\tilde{x}^{(k+1)}) \right) + k t_k \left( f^* - f(\tilde{x}^{(k+1)}) \right) - r\alpha_k D_2^{(k+1)} - r\beta_k D_1^{(k+1)} \\
&\quad + r \left\langle \nabla\psi(\tilde{z}^{(k+1)}) - \nabla\psi(\hat{z}^{(k+1)}), \tilde{z}^{(k+1)} - x^* \right\rangle \\
&= \frac{(2k+1-rk)t_k}{r} \left( f(\tilde{x}^{(k+1)}) - f^* \right) - \frac{k^2(t_{k-1} - t_k)}{r} \left( f(\tilde{x}^{(k)}) - f^* \right) - r\alpha_k D_2^{(k+1)} - r\beta_k D_1^{(k+1)} \\
&\quad + r \left\langle \nabla\psi(\tilde{z}^{(k+1)}) - \nabla\psi(\hat{z}^{(k+1)}), \tilde{z}^{(k+1)} - x^* \right\rangle.
\end{aligned}
$$

For the desired inequality to hold, it suffices that $\alpha_k, \beta_k \geq 0$. Recalling their definitions (and $\lambda_k$), we want

$$
\frac{1}{2L_{\psi^*}} - \frac{kL_R}{2(r+k)\gamma} \geq 0 \quad \text{and} \quad \frac{k(r+k)l_R}{2r^2\gamma} - \frac{L_f k^2 t_k}{2r^2} - \frac{L_f k t_k}{2r} \geq 0,
$$

which are equivalent to

$$
\gamma \geq \frac{kr}{kr + r^2} L_R L_{\psi^*} \quad \text{and} \quad t_k \leq \frac{l_R}{L_f \gamma}.
$$

It thus suffices to have

$$
\gamma \geq L_R L_{\psi^*}, \quad t_k \leq \frac{l_R}{L_f \gamma}.
$$

Under these conditions, we get the bound as required

$$
\begin{aligned}
\tilde{E}^{(k+1)} - \tilde{E}^{(k)} &\leq \frac{(2k+1-rk)t_k}{r} \left( f(\tilde{x}^{(k+1)}) - f^* \right) - \frac{k^2(t_{k-1} - t_k)}{r} \left( f(\tilde{x}^{(k)}) - f^* \right) \\
&\quad + r \left\langle \nabla\psi(\tilde{z}^{(k+1)}) - \nabla\psi(\hat{z}^{(k+1)}), \tilde{z}^{(k+1)} - x^* \right\rangle.
\end{aligned}
$$

$\square$

## A.2   Proof of Proposition 1

**Proposition 4.** *Assume the conditions as in Lemma 1. Then*

$$
\tilde{E}^{(1)} \leq r B_{\psi^*}(\nabla\psi(x^{(0)}), \nabla\psi(x^*)) + \frac{t_0}{r}(f(x^{(0)}) - f^*) + r \left\langle \nabla\psi(\tilde{z}^{(1)}) - \nabla\psi(\hat{z}^{(1)}), \tilde{z}^{(1)} - x^* \right\rangle \tag{35}
$$

*Proof.* From Lemma 1 applied with $k = 0$, we have

$$
\begin{aligned}
\tilde{E}^{(1)} &\leq \tilde{E}^{(0)} + \frac{t_0}{r} \left( f(\tilde{x}^{(1)}) - f^* \right) + r \left\langle \nabla\psi(\tilde{z}^{(1)}) - \nabla\psi(\hat{z}^{(1)}), \tilde{z}^{(1)} - x^* \right\rangle \\
&= r B_{\psi^*} \left( \nabla\psi(z^{(0)}), \nabla\psi(x^*) \right) + \frac{t_0}{r} \left( f(\tilde{x}^{(1)}) - f^* \right) + r \left\langle \nabla\psi(\tilde{z}^{(1)}) - \nabla\psi(\hat{z}^{(1)}), \tilde{z}^{(1)} - x^* \right\rangle.
\end{aligned}
$$

By definition, $\tilde{x}^{(1)} = \arg\min_{\tilde{x} \in \mathbb{R}^n} \gamma t_0 \langle \nabla f(x^{(1)}), \tilde{x} \rangle + R(\tilde{x}, x^{(1)})$, thus (since $R(x, x) = 0$)

$$
\gamma t_0 \left\langle \nabla f(x^{(1)}), \tilde{x}^{(1)} \right\rangle + R(\tilde{x}^{(1)}, x^{(1)}) \leq \gamma t_0 \left\langle \nabla f(x^{(1)}), x^{(1)} \right\rangle. \tag{36}
$$

Therefore, we get

$$
\begin{aligned}
f(\tilde{x}^{(1)}) - f^* &\leq \left\langle \nabla f(x^{(1)}), \tilde{x}^{(1)} - x^* \right\rangle + \frac{L_f}{2} \|\tilde{x}^{(1)} - x^{(1)}\|^2 && \text{by Lemma 3} \\
&\leq \left\langle \nabla f(x^{(1)}), \tilde{x}^{(1)} - x^* \right\rangle + \frac{L_f}{l_R} R(\tilde{x}^{(1)}, x^{(1)}) && \text{by assumption on } R \\
&\leq \left\langle \nabla f(x^{(1)}), \tilde{x}^{(1)} - x^* \right\rangle + \frac{1}{\gamma t_0} R(\tilde{x}^{(1)}, x^{(1)}) - \frac{L_f}{l_R} R(\tilde{x}^{(1)}, x^{(1)}) && \text{using } \frac{2L_f}{l_R} \leq \frac{1}{\gamma t_0} \\
&\leq \left\langle \nabla f(x^{(1)}), x^{(1)} - x^* \right\rangle - \frac{L_f}{l_R} R(\tilde{x}^{(1)}, x^{(1)}) && \text{using Equation 36} \\
&\leq f(x^{(1)}) - f^* + \frac{L_f}{2} \|x^{(1)} - x^*\|^2 - \frac{L_f}{l_R} R(\tilde{x}^{(1)}, x^{(1)}) && \text{using Lemma 3} \\
&\leq f(x^{(1)}) - f^* && \text{by assumption on } R.
\end{aligned}
$$

Now recalling that $x^{(1)} = \lambda_0 \tilde{z}^{(0)} + (1 - \lambda_0)\tilde{x}^{(0)}$ and $\lambda_0 = 1$, we have $x^{(1)} = \tilde{z}^{(0)} = x^{(0)}$. Therefore, we have $f(\tilde{x}^{(1)}) - f^* \leq f(x^{(0)}) - f^*$. Further using $\tilde{z}^{(0)} = x^{(0)}$ shows the desired inequality. $\qquad\square$

## B  Proof of Section 3 (approximate SMD)

The following lemma characterizes the effect of a mirror step on the Bregman divergence. We use this lemma to show an optimality gap bound on SMD with the inexact stochastic oracle $\mathsf{G}(x, \xi)$ similarly to Nemirovski et al. (2009), where the bound depends on the variance of the stochastic component $\Delta(x, \xi)$ and the norm of the inexactness $U(x)$.

**Lemma 6** (Nemirovski et al. 2009, Lem. 2.1). *For any $u, x \in \mathcal{X}$ and $y \in (\mathbb{R}^n)^*$, we have (where $\psi$ is $\alpha$-strongly convex),*

$$
B_\psi(u, P_x(y)) \leq B_\psi(u, x) + \langle y, u - x \rangle + \frac{\|y\|_*^2}{2\alpha}. \tag{37}
$$

Applying Equation 37 with $x = x^{(k)}$, $y = t_k \mathsf{G}(x^{(k)}, \xi^{(k)})$, given any point $u \in \mathcal{X}$, we have

$$
t_k \langle \mathsf{G}(x^{(k)}, \xi^{(k)}), x^{(k)} - u \rangle \leq B_\psi(u, x^{(k)}) - B_\psi(u, x^{(k+1)}) + \frac{t_k^2}{2\alpha} \|\mathsf{G}(x^{(k)}, \xi^{(k)})\|_*^2. \tag{38}
$$

Using the definition of $\mathsf{G}$, we can expand

$$
\begin{aligned}
t_k \langle g(x^{(k)}), x^{(k)} - u \rangle \leq\ & B_\psi(u, x^{(k)}) - B_\psi(u, x^{(k+1)}) + \frac{t_k^2}{2\alpha} \|\mathsf{G}(x^{(k)}, \xi^{(k)})\|_*^2 \\
& - t_k \langle \Delta_k, x^{(k)} - u \rangle - t_k \langle U_k, x^{(k)} - u \rangle.
\end{aligned} \tag{39}
$$

Observe that since $f$ is convex and $g(x) \in \partial f(x)$, we have $\langle g(x^{(k)}), x^{(k)} - u \rangle \geq f(x^{(k)}) - f(u)$. Using this, summing Equation 39 from 1 to $k$, and noting that $B_\psi \geq 0$, we have

$$
\sum_{i=0}^{k} t_i [f(x^{(i)}) - f(u)] \leq B_\psi(u, x^{(0)}) + \sum_{i=0}^{k} \frac{t_i^2}{2\alpha} \|\mathsf{G}(x^{(i)}, \xi^{(i)})\|_*^2 - \sum_{i=0}^{k} t_i \langle \Delta_i, x^{(i)} - u \rangle - \sum_{i=0}^{k} t_i \langle U_i, x^{(i)} - u \rangle. \tag{40}
$$

Observing that $\mathbb{E}[\Delta_i | \xi_0, ..., \xi_{i-1}] = 0$ and that $x^{(i)}$ lives in the $\sigma$-algebra defined by $\xi_0, ..., \xi_{i-1}$, we have that for any $i$,

$$
\mathbb{E}[\langle \Delta_i, x^{(i)} - u \rangle] = \mathbb{E}\left[ \mathbb{E}\left[ \langle \Delta_i, x^{(i)} - u \rangle \mid \xi_0, ..., \xi_{i-1} \right] \right] = 0.
$$

We can take expectations over $\xi$ to get

$$
\mathbb{E}\left[ \sum_{i=0}^{k} t_i [f(x^{(i)}) - f(u)] \right] \leq B_\psi(u, x^{(0)}) + \sum_{i=0}^{k} \frac{t_i^2}{2\alpha} \mathbb{E}\left[ \|\mathsf{G}(x^{(i)}, \xi^{(i)})\|_*^2 \right] - \sum_{i=0}^{k} t_i \mathbb{E}\left[ \langle U_i, x^{(i)} - u \rangle \right].
$$

Assuming that the stochasticity is bounded

$$\mathbb{E}[\|\mathsf{G}(x,\xi)\|_*^2] \le \sigma^2 \quad \forall x \in \mathcal{X},$$

this gives an optimality gap bound, which can be extended to convergence of the ergodic average.

If $f$ were $\mu$-strongly convex, then we can remove the uniform boundedness assumption since this allows us to control $\|x^{(i)} - u\|$, using the fact that $\langle g(x^{(k)}), x^{(k)} - u \rangle \ge f(x^{(k)}) - f(u) + \mu \|x^{(k)} - u\|^2/2$. Equation 40 instead becomes

$$\sum_{i=0}^{k} t_i[f(x^{(i)}) - f(u)] \le B_\psi(u, x^{(0)}) + \sum_{i=0}^{k} \frac{t_i^2}{2\alpha} \|\mathsf{G}(x^{(i)}, \xi^{(i)})\|_*^2$$
$$- \sum_{i=0}^{k} t_i \langle \Delta_i, x^{(i)} - u \rangle - \sum_{i=0}^{k} t_i \left[ \langle U_i, x^{(i)} - u \rangle + \frac{\mu}{2} \|x^{(i)} - u\|^2 \right]. \quad (41)$$

Taking expectations as before and using Young's inequality on the final term, we get

$$\mathbb{E}\left[ \sum_{i=0}^{k} t_i[f(x^{(i)}) - f(u)] \right] \le B_\psi(u, x^{(0)}) + \sum_{i=0}^{k} \frac{t_i^2}{2\alpha} \mathbb{E}\left[ \|\mathsf{G}(x^{(i)}, \xi^{(i)})\|_*^2 \right] + \sum_{i=0}^{k} \frac{t_i}{2\mu} \|U_i\|_*^2.$$

## C   Proof of Theorem 3 (approximate ASMD)

**Theorem 4.** *Suppose $f$ has $L_f$-Lipschitz gradient, the mirror map is $\alpha$-strongly convex, the approximation error $U^{(k)}$ is bounded, and that the stochastic oracle is otherwise unbiased with bounded second moments. Assume the diameter of $\mathcal{X}$ is finite, that is, there exists a constant $M_\psi > 0$ such that*

$$M_\psi = \sup_{x, x' \in \mathcal{X}} B_\psi(x, x').$$

*Suppose further that there exists a constant $K$ such that for every iterate $x^{(k)}$, we have*

$$\langle \nabla f(x^{(k)}), U^{(k)} \rangle \le K.$$

*Then the convergence rate of approximate ASMD is (where the right hand side is also given by Equation 44),*

$$\mathbb{E}[f(x^{(k)}) - f(x^*)] \le K + \mathcal{O}(k^{-2} + k^{-1/2}).$$

*Proof.* Recall the definition of the energy,

$$\mathcal{E}^{(k)} = \mathbb{E}[A_k(f(x^{(k)}) - f(x^*)) + s_k B_\psi(x^*, \nabla \psi^*(y^{(k)}))]. \quad (42)$$

Using the following identity (which can be shown by expanding the Bregman distances using their definitions and using $\nabla \psi^* = (\nabla \psi)^{-1}$)

$$B_\psi(x^*, \nabla \psi^*(y^{(k+1)})) - B_\psi(x^*, \nabla \psi^*(y^{(k)})) - B_\psi(\nabla \psi^*(y^{(k)}), \nabla \psi^*(y^{(k+1)})) = \langle y^{(k)} - y^{(k+1)}, x^* - \nabla \psi^*(y^{(k)}) \rangle,$$

we compute the difference in energy

$$\begin{aligned}
\mathcal{E}^{(k+1)} - \mathcal{E}^{(k)} &= \mathbb{E}[A_{k+1}(f(x^{(k+1)}) - f(x^*)) - A_k(f(x^{(k)}) - f(x^*))] + \mathbb{E}[(s_{k+1} - s_k)B_\psi(x^*, \nabla \psi^*(y^{(k+1)}))] \\
&\quad + s_k \mathbb{E}[B_\psi(x^*, \nabla \psi^*(y^{(k+1)})) - B_\psi(x^*, \nabla \psi^*(y^{(k)}))] \\
&= \mathbb{E}[A_{k+1}(f(x^{(k+1)}) - f(x^*)) - A_k(f(x^{(k)}) - f(x^*))] + \mathbb{E}[(s_{k+1} - s_k)B_\psi(x^*, \nabla \psi^*(y^{(k+1)}))] \\
&\quad + s_k \mathbb{E}[B_\psi(\nabla \psi^*(y^{(k)}), \nabla \psi^*(y^{(k+1)})) + \langle y^{(k)} - y^{(k+1)}, x^* - \nabla \psi^*(y^{(k)}) \rangle] \\
&= \mathbb{E}[A_{k+1}(f(x^{(k+1)}) - f(x^*)) - A_k(f(x^{(k)}) - f(x^*))] + \mathbb{E}[(s_{k+1} - s_k)B_\psi(x^*, \nabla \psi^*(y^{(k+1)}))] \\
&\quad + \mathbb{E}[s_k B_\psi(\nabla \psi^*(y^{(k)}), \nabla \psi^*(y^{(k+1)})) + (A_{k+1} - A_k)\langle \mathsf{G}(x^{(k+1)}, \xi^{(k+1)}), x^* - \nabla \psi^*(y^{(k)}) \rangle]
\end{aligned}$$

where the second equality comes from the above identity, and the third equality from the definition of $y^{(k)}$. We further compute using the optimality conditions in Equations (18) and (19) and expanding $\mathsf{G}(x^{(k+1)}, \xi^{(k+1)}) = \nabla f(x^{(k+1)}) + \Delta_{k+1}$:

$$
\begin{aligned}
\mathcal{E}^{(k+1)} - \mathcal{E}^{(k)} &= \mathbb{E}[A_{k+1}(f(x^{(k+1)}) - f(x^*)) - A_k(f(x^{(k)}) - f(x^*))] + \mathbb{E}[(s_{k+1} - s_k)B_\psi(x^*, \nabla\psi^*(y^{(k+1)}))] \\
&\quad + \mathbb{E}[s_k B_\psi(\nabla\psi^*(y^{(k)}), \nabla\psi^*(y^{(k+1)}))] \\
&\quad + \mathbb{E}[(A_{k+1} - A_k)\langle \nabla f(x^{(k+1)}), x^* - x^{(k+1)} \rangle - A_k\langle \nabla f(x^{(k+1)}), x^{(k+1)} - x^{(k)} \rangle] \\
&\quad + \mathbb{E}[(A_{k+1} - A_k)\langle \Delta_{k+1}, x^* - \nabla\psi^*(y^{(k)}) \rangle] \\
&\quad + \mathbb{E}[(A_{k+1} - A_k)\langle \mathsf{G}(x^{(k+1)}, \xi^{(k+1)}), U^{(k)} \rangle] \\
&\leq \mathbb{E}[A_{k+1}(f(x^{(k+1)}) - f(x^*)) - A_k(f(x^{(k)}) - f(x^*))] + \mathbb{E}[(s_{k+1} - s_k)B_\psi(x^*, \nabla\psi^*(y^{(k+1)}))] \\
&\quad + \mathbb{E}[s_k B_\psi(\nabla\psi^*(y^{(k)}), \nabla\psi^*(y^{(k+1)}))] \\
&\quad + \mathbb{E}[(A_{k+1} - A_k)(f(x^*) - f(x^{(k+1)})) + A_k(f(x^{(k)}) - f(x^{(k+1)})))\rangle] \\
&\quad + \mathbb{E}[(A_{k+1} - A_k)\langle \Delta_{k+1}, x^* - \nabla\psi^*(y^{(k)}) \rangle] \\
&\quad + \mathbb{E}[(A_{k+1} - A_k)\langle \mathsf{G}(x^{(k+1)}, \xi^{(k+1)}), U^{(k)} \rangle] \\
&= \mathbb{E}[(s_{k+1} - s_k)B_\psi(x^*, \nabla\psi^*(y^{(k+1)}))] + \mathbb{E}[s_k B_\psi(\nabla\psi^*(y^{(k)}), \nabla\psi^*(y^{(k+1)}))] \\
&\quad + \mathbb{E}[(A_{k+1} - A_k)\langle \nabla f(x_{k+1}), U^{(k)} \rangle]
\end{aligned}
$$

where the second inequality follows from convexity of $f$, and the final equality from zero mean of $\Delta_{k+1}$ and independence with $y^{(k)}, U^{(k)}$.

Recalling that since $\psi$ is $\alpha$-strongly convex, we have that $\psi^*$ is $1/\alpha$-strongly smooth. Using this, we have the following bound on the Bregman divergence:

$$
\begin{aligned}
B_\psi(\nabla\psi^*(y^{(k)}), \nabla\psi^*(y^{(k+1)})) = B_{\psi^*}(y^{(k+1)}, y^{(k)}) &\leq \frac{1}{2\alpha}\|y^{(k+1)} - y^{(k)}\|_*^2 \\
&\leq \frac{8L_f^2 M_\psi + 2\alpha\sigma^2 + 4\|\nabla f(x^*)\|_*^2}{\alpha} \frac{(A_{k+1} - A_k)^2}{2\alpha s_k^2},
\end{aligned}
$$

where the final iterate comes from the fact that for any $x \in \mathcal{X}$, $\|\nabla f(x)\|_* \leq \frac{L_f\sqrt{2M_\psi}}{\sqrt{\alpha}} + \|\nabla f(x^*)\|_*$, which is derived from the smoothness of $f$, strong convexity of $\psi$ and bounded domain assumption. Furthermore, we have $B_\psi(x^*, \nabla\psi^*(y^{(k+1)})) \leq M_\psi$. Further assume that $\langle \nabla f(x^{(k+1)}), U^{(k)} \rangle$ is bounded, say by $K$. Substituting back we have

$$
\mathcal{E}^{(k+1)} - \mathcal{E}^{(k)} \leq M_\psi(s_{k+1} - s_k) + \frac{4L_f^2 M_\psi + \alpha\sigma^2 + 2\|\nabla f(x^*)\|_*^2}{\alpha^2} \frac{(A_{k+1} - A_k)^2}{s_k} + |A_{k+1} - A_k|K.
$$

Assuming that $A_k$ is monotonically increasing (which will be justified later), we can sum from 0 to $k - 1$ and get

$$
\mathcal{E}^{(k)} \leq \mathcal{E}^{(0)} + M_\psi s_k + \frac{4L_f^2 M_\psi + \alpha\sigma^2 + 2\|\nabla f(x^*)\|_*^2}{\alpha^2} \sum_{j=0}^{k-1} \frac{(A_{j+1} - A_j)^2}{s_j} \\
+ A_k K.
$$

Taking $A_j = j(j+1)/2$ (which is monotonically increasing) and $s_j = (j+1)^{3/2}$, we get

$$
\mathcal{E}^{(k)} \leq \mathcal{E}^{(0)} + M_\psi(k+1)^{3/2} + \frac{4L_f^2 M_\psi + \alpha\sigma^2 + 2\|\nabla f(x^*)\|_*^2}{\alpha^2} \sum_{j=0}^{k-1} \sqrt{j+1} + A_k K
$$

$$
\leq \mathcal{E}^{(0)} + M_\psi(k+1)^{3/2} + \frac{8L_f^2 M_\psi + 2\alpha\sigma^2 + 4\|\nabla f(x^*)\|_*^2}{\alpha^2}(k+1)^{3/2} + A_k K.
$$

where we use the following lemma for divergent series for the final inequality.

**Lemma 7** (Chlebus 2009). *For $p < 0$,*

$$1 + \frac{k^{1-p} - 1}{1 - p} \leq \sum_{j=1}^{k} j^{-p} \leq \frac{(k+1)^{1-p} - 1}{1 - p} \tag{43}$$

Dividing throughout by $A_k$, we get

$$\mathbb{E}[f(x^{(k)}) - f(x^*)] \leq \mathcal{E}^{(0)}/A_k + \frac{(k+1)^{3/2}}{A_k} \left[ M_\psi + \frac{8L_f^2 M_\psi + 2\alpha\sigma^2 + 4\|\nabla f(x^*)\|_*^2}{\alpha^2} \right] + K \tag{44}$$

$$= K + \mathcal{O}(k^{-2} + k^{-1/2}).$$

$\square$

# D  Proofs in Section 6

## D.1  Proof of Proposition 2

**Proposition 2.** *If $\Omega_0$ is gradient descent on the $G$-invariant objective function $L$, then it is $G$-equivariant, i.e. $\Omega_0(g \cdot z) = g \cdot [\Omega_0(z)]$ for all $g \in G, z \in \mathcal{Z}$.*

*Proof.* We first check that $\nabla L$ commutes with $g$ actions. For any $y \in \mathcal{Z}$, the derivative operator $DL : \mathcal{Z} \to \mathcal{Z}^*$ satisfies

$$
\begin{aligned}
DL(g \cdot z)(y) &= \lim_{t \to 0} \frac{L(g \cdot z + ty) - L(g \cdot z)}{t} \\
&= \lim_{t \to 0} \frac{L(z - tg^{-1} \cdot y) - L(z)}{t} \\
&= DL(z)(g^{-1}y),
\end{aligned}
$$

where the second inequality holds since $L(g \cdot z) = L(z)$, applied with $g^{-1}$ and linearity of $g$ actions. Let $\iota$ denote the canonical isomorphism $\iota : \mathcal{Z}^* \to \mathcal{Z}$. For any $y \in \mathcal{Z}$,

$$
\begin{aligned}
\langle \iota^{-1} \nabla L(g \cdot z), y \rangle &= DL(g \cdot z)(y) \\
&= DL(z)(g^{-1} \cdot y) \\
&= \langle \iota^{-1} \nabla L(z), g^{-1} \cdot y \rangle \\
&= \langle g \cdot \iota^{-1} \nabla L(z), y \rangle.
\end{aligned}
$$

Hence, $G$ commutes with $\iota^{-1}\nabla L = DL$. We now have that

$$
\begin{aligned}
\Omega_0(g \cdot z) &= g \cdot z - \eta(\nabla L)(g \cdot z) \\
&= g \cdot z - \eta g \cdot \nabla L(z) \\
&= g \cdot (z - \eta g \nabla L(z)) \\
&= g \cdot \Omega_0(z),
\end{aligned}
$$

where the first equality comes from the fact that $g$ trivially commutes with $\iota$ (and thus $\iota^{-1}$) using the induced group action, and the second and third equality comes from the linearity of $g$ actions. Since this is true for all $y \in \mathcal{Z}$, by the Riesz representation theorem, we have $\nabla L(g \cdot z) = g \cdot \nabla L(z)$. $\square$

# E  Step-size comparison for baseline methods

Here we detail the step-sizes used to compare. In general, we choose the best performing method at 100 iterations, with preference given to the one converging to a smaller value at higher iterations in case of ambiguity. Note later iteration cutoffs usually mean insignificant performance at earlier iterations. The grid used for the step-sizes are of the form $\{1, 2, 5\} \times 10^{-k}$. The endpoints are

- **GD** $[2 \times 10^{-4}, 1 \times 10^{-1}]$

- **Adam** $[1 \times 10^{-3}, 1 \times 10^{-1}]$

- **Nesterov** $[1 \times 10^{-4}, 5 \times 10^{-3}]$

Figure 9 shows the loss evolution for the proposed step-sizes up to 5000 iterations. We note that the grid is sufficiently wide such that smaller or larger step-sizes are suboptimal, or lead to slow convergence in the low iteration count setting. The specific choices are as below.

**Image denoising.** GD: $1 \times 10^{-2}$. Adam: $2 \times 10^{-3}$. Nesterov: $1 \times 10^{-3}$.

**Image inpainting.** GD: $1 \times 10^{-2}$. Adam: $1 \times 10^{-2}$. Nesterov: $2 \times 10^{-3}$.

**SVM training.** GD: $1 \times 10^{-2}$. Adam: $2 \times 10^{-2}$. Nesterov: $5 \times 10^{-3}$.

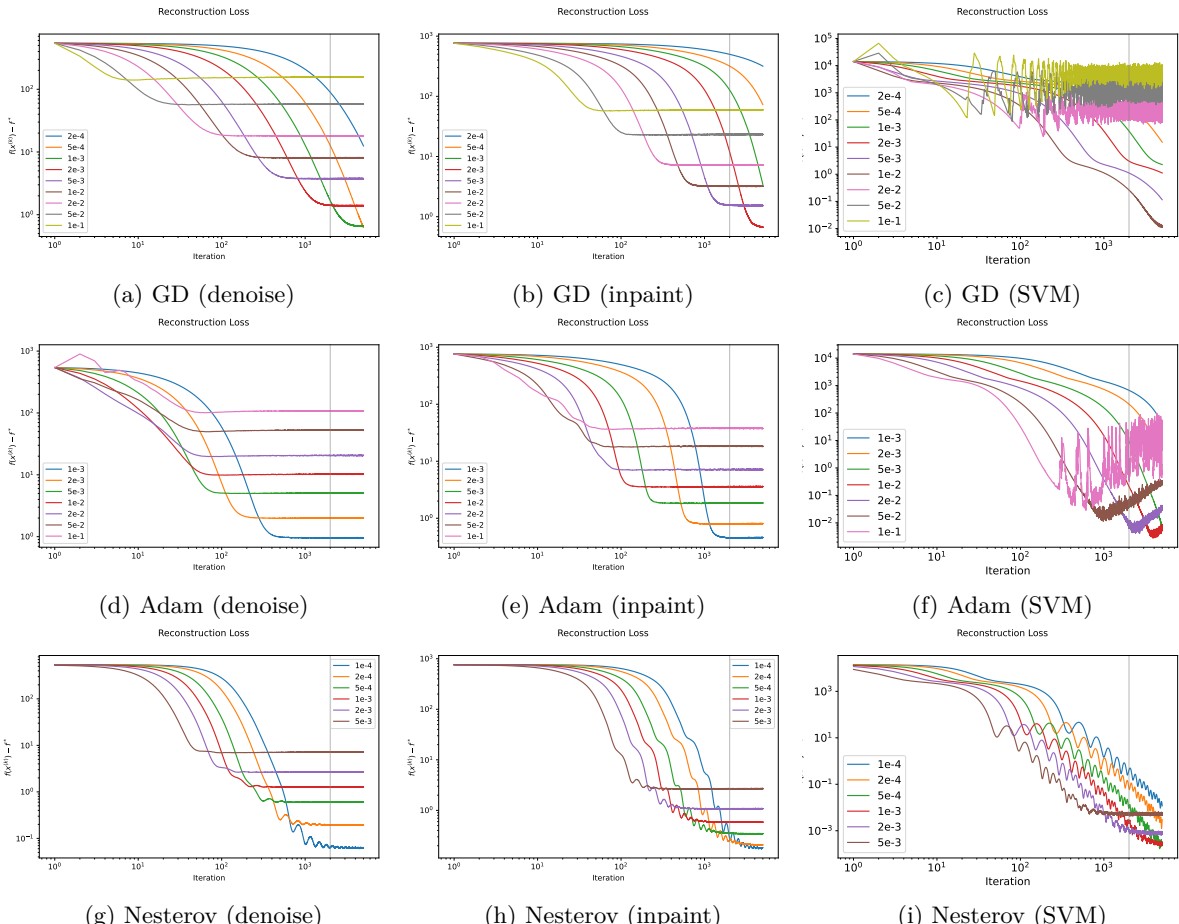

Figure 9: Comparison of the baseline methods with various step-sizes for the ellipse denoising and image inpainting tasks. Vertical gray line represents iteration 2000, which is the maximum iteration considered in the main text figures.

# F   Comparison of loss against time

We consider the image inpainting experiment and plot against the wall-clock time instead of the iteration count. In Figure 10, we consider various different step-sizes for the baselines GD, Nesterov accelerated GD, and Adam, as chosen in Appendix E. Each solid line corresponds to a different choice of step-size. We observe

that the baseline methods plateau after a period of fast optimization, while the proposed LAMD method is able to continue decreasing.

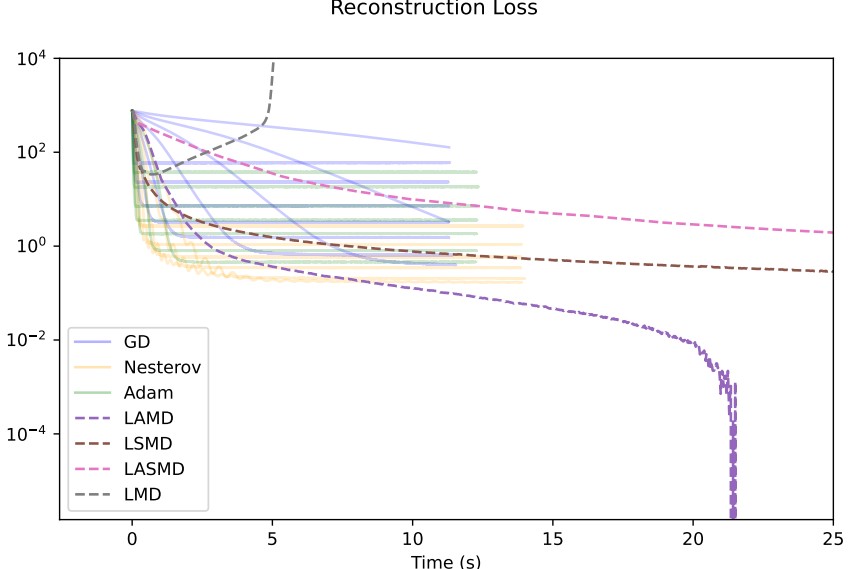

Figure 10: Plot of reconstruction loss against wall-clock time for image inpainting experiment.

In Figure 11, we plot the training loss of CNN training on MNIST as in Section 7.2. Due to the compact parameterization, we observe that the diagonal LMD methods introduce minor overheads compared to the baseline SGD and Adam methods, which have been optimized in the PyTorch framework, retaining its competitive nature when plotting against iteration count.

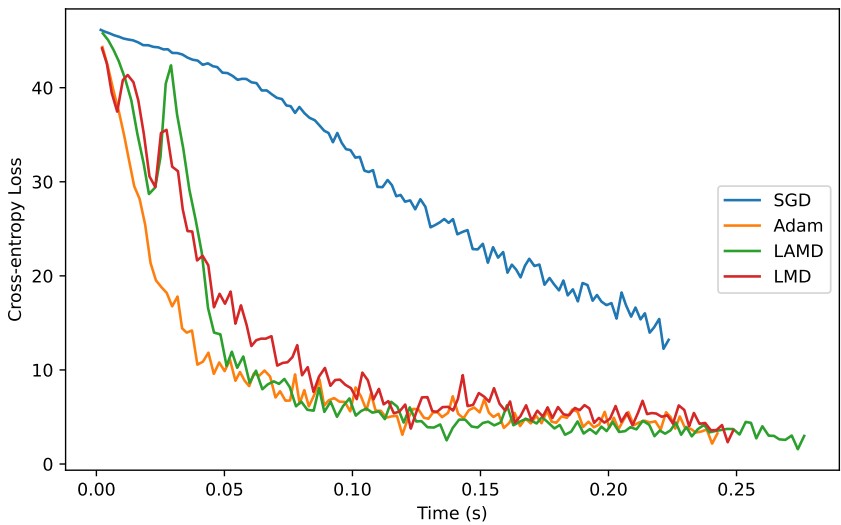

Figure 11: Plot of training cross-entropy loss against wall-clock time when training CNN on MNIST. The wall-clock time includes only the optimization phases and excludes overhead resulting from testing at each iteration.

## G   Monotonic spline parameterization

In Section 7, we utilize an element-wise spline-based parameterization (shared across parameters of the same layer) of the mirror map. Here we detail the implementation of the spline.

Recall that we define the mirror map (acting element-wise) to be

$$\psi : \mathbb{R} \to \mathbb{R}, \ \psi(x) = \int_0^x \sigma(t)\mathrm{d}t.$$

Here, $\sigma$ is a learnable piecewise-linear monotonic spline, with $\sigma(0) = 0$. Thus $\psi(x)$ is a piecewise quadratic monotonic spline. $\sigma$ can be parameterized in terms of knot locations $t_{-K} < ... < t_0 = 0 < ... < t_K$, as well as the values $c_i = \sigma(t_i)$, with linear interpolation between the intervals (and extrapolation on $(-\infty, t_{-K})$ and $(t_K, +\infty)$). The monotonic condition can be enforced during training by forcing $c_{-K} < c_{-K+1} < ... < c_K$. We note that shifting $\sigma$ up or down by a constant does not change the mirror map dynamics, so without loss of generality $c_0$ can be taken to be 0.

$$\sigma(x) = \begin{cases} c_i + \frac{x - t_i}{t_{i+1} - t_i}(c_{i+1} - c_i) & , x \in [t_i, t_{i+1}]; \\ c_{-K+1} - \frac{t_{-K+1} - x}{t_{-K+1} - t_K}(c_{-K+1} - c_{-K}) & , x < t_{-K}; \\ c_{K-1} + \frac{x - t_{K-1}}{t_K - t_{K-1}}(c_K - c_{K-1}) & , x > t_K. \end{cases}$$

In our implementation, we fix the knots as in Goujon et al. (2022) to be $t_i = 0.05i$, $i = -20, ..., 20$, which reduces the number of learnable parameters to only the values of $\sigma$ at the knots, which is $2K = 40$. One useful point of this characterization is that the inverse of $\sigma$ can be easily computed by computing the values at the knots $\sigma(t_i)$ and linearly interpolating.

## H   Reptile Initialization for Section 7.3

In Section 7.3, we compared LMD with baseline methods from a meta-learning objective. Informally, Reptile can be treated as an algorithm that finds a solution that is close to each task's manifold of optimal solutions. Here we detail how the Reptile initialization is computed for our purpose. The algorithm is as follows:

---

**Algorithm 8** Reptile (Nichol et al., 2018, Alg. 1)

---

1: Initialize parameters $\theta$
2: **for** $k = 1, ..., K$ **do**
3:     Sample task $f \in \mathcal{F}$
4:     Perform $n$ steps of SGD/Adam on $f$ from $\theta$ to obtain parameters $\tilde{\theta}$
5:     Update initialization $\theta \leftarrow \theta + \epsilon(\tilde{\theta} - \theta)$
6: **end for**
7: **return** initialization $\theta$

---

We consider training on the 5-shot 10-way Omniglot dataset for $K = 5 \times 10^5$ meta-epochs. The inner optimization loop in Step 4 is done using full-batch SGD with learning rate $1 \times 10^{-2}$ for $n = 10$ iterations. The meta-stepsize $\epsilon$ is decreased linearly from $\epsilon = 0.1$ to 0 over the entire training process. This returns the Reptile initialization $\theta_K$.

