# OpenReview forum: "Boosting Data-Driven Mirror Descent with Randomization, Equivariance, and Acceleration"
_TMLR — Accepted by TMLR_

### Review · Reviewer_QBq1 · 2024-02-18

**Summary Of Contributions:**

This paper studies the problem of mirror descent optimization in the case of the distance generating functions in the Bregman divergences being learned. Building upon past work that analyzes the convergence of full batch mirror descent in the case of approximate mirror maps, they show that under a good-enough approximation an accelerated variant will converge to the optimal value. They further extend these results to the stochastic setting with unbounded errors, and conduct several empirical studies in convex and non-convex settings.

**Audience:**

Yes

**Claims And Evidence:**

No

**Requested Changes:**

1. [critical] Substantively addressing the first two weaknesses above, namely (1) explaining why the assumption in Equation 12 is reasonable or why a more natural assumption would not work and (2) addressing related theoretical work, especially that related to learned mirror descent. I believe the first point is critical for establishing convincing evidence of the authors' claims and the second point is critical for clarifying the claims in comparison with past work.
2. While not critical due to the TMLR guidelines, more substantive empirical evaluations, in terms of benchmarks and baselines, in the neural network setting would significantly strengthen the work.

**Strengths And Weaknesses:**

## Strengths:
1. While the idea of analyzing approximate mirror descent in order to justify learned mirror maps is not new, the result showing that acceleration suffices to converge to the true optimum is an interesting advance upon past work.
2. The authors take seriously the question of scaling up the algorithms using stochastic optimization rather than full-batch optimization, as well as for reducing the number of parameters on larger-scale (neural network) problems.
3. Empirically, the problem of amortizing optimization cost by learning an optimizer across many optimization tasks seems well-motivated, and the authors demonstrate improved performance of their approach over baselines/past unaccelerated methods on some of the convex problems.

## Weaknesses:
1. Theoretically, it is unclear to me how reasonable the Theorem 1 assumption given by equation (12) is. While some bound on the error of the approximate mirror map is necessary, the specific bound used here assumes the algorithmic trajectories of the approximate and non-approximate versions are close across all iterations of the algorithm. Conditions under which this holds are not given, and it is unclear to me why the assumption is somehow more natural or simple than the (un-interesting) approach of just assuming the result of the theorem directly. A more natural assumption seems to me some type of functional bound on the potentials $M_\varphi$ and $M_\theta$ themselves.
2. There is a large amount of missing related work in the theory community. Learned mirror descent, including the Bregman divergence, has been studied in the online setting by Denevi et al. (2019) and Khodak et al. (2019), as well as in the statistical setting by Gao et al. (2022). There is a variety of other related theoretical work—although not regarding mirror descent explicitly—under the names “amortized optimization” (Amos, 2023), “meta-optimization” (Chen & Hazan, 2023), and “data-driven algorithm design” (Gupta & Roughgarden, 2017). While the theory presented here is distinct, comparison (both theoretical and possibly empirical) is warranted.
3. The empirical results are somewhat underwhelming overall, especially in the neural network case. The meta-learned optimizer does not significantly outperform Adam—despite presumably much greater time spent due to meta-optimization—and there is no comparison to standard benchmarks in the learning-to-optimize literature such as Omniglot (Lake et al., 2011) and Mini-ImageNet (Vinyals, 2016). Would the proposed (fairly-involved) approaches outperform basic initialization-learning methods like MAML (Finn et al., 2017) or Reptile (Nichol et al., 2018) on such benchmarks? If such few-shot learning benchmarks are not the target application, what deep learning application is, beyond MNIST?
4. The paper is extremely lengthy and poorly organized. While it is the authors’ decision, it seems to me the theoretical arguments could largely be moved to the appendix, as they do not seem to significantly factor into the main message of the paper.

## References:
1. Amos. *Tutorial on amortized optimization*. FTML 2023.
2. Chen, Hazan. *A nonstochastic control approach to optimization*. NeurIPS 2023.
3. Denevi, Stamos, Ciliberto, Pontil. *Online-within-online meta-learning*. NeurIPS 2019.
4. Finn, Abbeel, Levine. *Model-agnostic meta-learning for fast adaptation of deep networks*. ICML 2017.
5. Gao, Gouk, Lee, Hospedales. *Meta mirror descent: Optimiser learning for fast convergence*. 2022.
6. Gupta, Roughgarden. *A PAC approach to application-specific algorithm selection*. SIAM J. Computing, 2017.
7. Khodak, Balcan, Talwalkar. *Adaptive gradient-based meta-learning methods*. NeurIPS 2019.
8. Lake, Salakhutdinov, Gross, Tenenbaum. *One-shot learning of simple visual concepts*. CogSci, 2011.
9. Nichol, Achiam, Schulman. *On first-order meta-learning methods*. 2018.
10. Vinyals, Blundell, Lillicrap, Kavukcuoglu, Wierstra. *Matching networks for one shot learning*. NeurIPS 2016.

---

> ### Author Response · Authors · 2024-03-20
> **Rebuttal to Reviewer QBq1**
>
> We thank the reviewer for the detailed and timely feedback. Please find below point-by-point responses to the reviewer's concerns.
>
> > Theoretically, it is unclear to me how reasonable the Theorem 1 assumption given by equation (12) is. While some bound on the error of the approximate mirror map is necessary, the specific bound used here assumes the algorithmic trajectories of the approximate and non-approximate versions are close across all iterations of the algorithm. Conditions under which this holds are not given, and it is unclear to me why the assumption is somehow more natural or simple than the (un-interesting) approach of just assuming the result of the theorem directly. A more natural assumption seems to me some type of functional bound on the potentials $M_\varphi$ and $M_\theta$ themselves.
>
> We note that Equation (12) (now Equation 8) is the most general assumption possible, and only assumes an upper bound on the error term. Indeed, if the approximation error residual happens to point away from the minimizer, then the term in Equation (8) will be negative, and the energy $\tilde{E}^{(k)}$ will increase slower than the worst case of $\tilde{E}^{(1)}+krM$, leading to faster empirical convergence.
>
> &nbsp;&nbsp;&nbsp;&nbsp;Nonetheless, consider the case of LMD, where $\psi = M_\theta$, the approximate MD iterate is $\tilde{z}^{(k+1)} = \nabla M_\vartheta^* (\nabla M_\theta \tilde{z}^{(k)} - t_k \nabla f( \tilde{z}^{(k)}))$, and the true MD iterate is $\hat{z}^{(k+1)} = \nabla M_\theta^{-1} (\nabla M_\theta \tilde{z}^{(k)} - t_k \nabla f(\tilde{z}^{(k)}))$. A simple relaxation of the assumption is that the approximation error $\|\nabla M_\theta \circ \nabla M_\vartheta^* - I\|$ is uniformly bounded in the dual space on the dual points $\tilde{z}^{(k)} - t_k \nabla f( \tilde{z}^{(k)})$ (i.e. ``$M_\theta$ is a good inverse mirror map to $M_\vartheta^*$"). By Cauchy-Schwarz, this bounds the LHS of the product in (8), and with a bounded iterates assumption, is sufficient to give a uniform bound on the term in Equation (8). Similar relaxations are given after Theorem 2 and Theorem 3, and now have been added to Theorem 1.
>
> &nbsp;&nbsp;&nbsp;&nbsp;As mentioned above by the reviewer, the same assumption used for approximate MD now allows for convergence to the minimizer when used for approximate accelerated MD. We note that this is necessary, as empirical results have shown that LMD indeed only converges to a constant above the minimum (if not diverging).
>
> >There is a large amount of missing related work in the theory community. Learned mirror descent, including the Bregman divergence, has been studied in the online setting by Denevi et al. (2019) and Khodak et al. (2019), as well as in the statistical setting by Gao et al. (2022). There is a variety of other related theoretical work—although not regarding mirror descent explicitly—under the names “amortized optimization” (Amos, 2023), “meta-optimization” (Chen & Hazan, 2023), and “data-driven algorithm design” (Gupta & Roughgarden, 2017). While the theory presented here is distinct, comparison (both theoretical and possibly empirical) is warranted.
>
> We thank the reviewer for the theoretical references, as well as informing us of the Gao et al. paper, which appears to be a diagonal quadratic version of mirror descent. We have added a couple of additional paragraphs in the introduction to incorporate and contrast the previous statistical theory compared to the convex theory that is already presented in the introduction.

---

> > ### Author Response · Authors · 2024-03-20
> > **Followup to Rebuttal**
> >
> > > The empirical results are somewhat underwhelming overall, especially in the neural network case. The meta-learned optimizer does not significantly outperform Adam—despite presumably much greater time spent due to meta-optimization—and there is no comparison to standard benchmarks in the learning-to-optimize literature such as Omniglot (Lake et al., 2011) and Mini-ImageNet (Vinyals, 2016). Would the proposed (fairly-involved) approaches outperform basic initialization-learning methods like MAML (Finn et al., 2017) or Reptile (Nichol et al., 2018) on such benchmarks? If such few-shot learning benchmarks are not the target application, what deep learning application is, beyond MNIST?
> >
> > The LMD (and extensions) method has been developed for convex problems, and the first set of empirical results is intended to demonstrate the convergence behavior compared to commonly used and historically effective methods. Figures 3 and 4 additionally demonstrate that Adam does not converge to a minimizer, whereas LAMD does as suggested by Theorem 1. The intention of the neural network experiments is to demonstrate that even though the method and theory is developed for the convex case, with some small modifications, the method can also be utilized for classes of non-convex problems.
> >
> > &nbsp;&nbsp;&nbsp;&nbsp;Regarding the time needed to use the meta-learned optimizer, we have added Appendix F that plots the training objective against wall-clock time for the image inpainting and CNN training experiments. We observe better convergence for LAMD in the inpainting setting, and only minor increase in computation time for the MNIST CNN experiment. We note that the time to train LMD is around the same as Reptile for the Omniglot meta-learning task.
> >
> > &nbsp;&nbsp;&nbsp;&nbsp;We would like to clarify that Section 6 on equivariant MD is intended as general theory for L2O, and is not restricted to LMD. This can be used as a first step towards deep learning applications of LMD and a justification of existing layer-wise mirror maps. The mirror maps have to be exact inverse of each other due to the sensitivity/high Lipschitz constant of the neural network training problem. Therefore, it is to be expected that the current LMD may not be able to outperform Adam, which may be due to a suboptimal parameterization (indeed, the Gao et al. paper mentioned in the previous point does not outperform Adam with properly tuned hyperparameters either).
> >
> > &nbsp;&nbsp;&nbsp;&nbsp;We would like to highlight that the Reptile/MAML appears to take a different approach to meta-learning as compared to L2O (as now added in the introduction). The meta-learning objective of Reptile/MAML appears to be find a good initialization in the sense that training is fast in any direction (where test-time adaptation is allowed), whereas the provable model-based L2O schemes such as LMD attempt to change the optimization path from any given initialization (without test-time adaptation). These are two different sides of the same objective.
> >
> > > The paper is extremely lengthy and poorly organized. While it is the authors’ decision, it seems to me the theoretical arguments could largely be moved to the appendix, as they do not seem to significantly factor into the main message of the paper.
> >
> > As suggested, we have moved the proofs of the approximate MD variants of Sections 2-4 to the appendix. We believe that the exposition in Section 6 illustrates the types of algebraic manipulations needed to show equivariance, and thus have only moved the proof of Proposition 2 (equivariance of gradient descent) to the appendix.
> >
> > ----
> > > [critical] Substantively addressing the first two weaknesses above, namely (1) explaining why the assumption in Equation 12 is reasonable or why a more natural assumption would not work and (2) addressing related theoretical work, especially that related to learned mirror descent. I believe the first point is critical for establishing convincing evidence of the authors' claims and the second point is critical for clarifying the claims in comparison with past work.
> >
> > We have addressed (1) by adding a functional interpretation of Equation (12) (now Eq. 8) in Remark 1 after Theorem 1, clarifying that a ``close-enough" pair of mirror maps and bounded iterates is sufficient to satisfy boundedness of Eq (8). We have additionally added some literature review on theoretical L2O/meta-learning in the introduction as suggested by the reviewer.
> >
> > > While not critical due to the TMLR guidelines, more substantive empirical evaluations, in terms of benchmarks and baselines, in the neural network setting would significantly strengthen the work.
> >
> > We have added experiments on Omniglot using Reptile for the neural network experiments section. We have also added a comparison with a learned primal-dual scheme of Banert et al. for the convex imaging experiments, where a proximal formulation is available.

---

> ### Comment · Reviewer_QBq1 · 2024-04-10
> **Response**
>
> Thank you for the updates. Some further notes:
> 1. I can roughly buy the justification for the result outlined in Remark 1, although the result would be stronger if the boundedness of iterates was guaranteed rather than assumed. However, it is not obvious to me why the statement starting with "the LHS is simply" is true. Does it not at least require assuming that $f$ has zero gradient at optimality? Writing out the identity explicitly would be useful. Moreover, presumably one should write $\nabla M_\theta\circ\nabla M_\vartheta^\ast$ instead of $\nabla M_\theta\nabla M_\vartheta^\ast$ since these are vector-valued functions and multiplying them doesn't make sense.
> 2. It is not clear to me whether $\hat z^{(k)}$ refers to the iterates resulting from running "true" mirror descent for the entire trajectory or just the sequence resulting from applying the "true" mirror descent update to $\tilde z^{(k-1)}$. I believe it is the latter, but this should be made more clear (in particular, I'm not sure if they should be described as "iterates" since one does not follow from the other; it is instead an auxiliary sequence associated with the approximate mirror descent iterates $\tilde z^{(k)}$).
> 3. I would suggest *not* using the word "regret" to describe any result in this paper or the results of Beck & Teboulle (2003), since its meaning is defined using the setup in online learning rather than optimization. A better word might be "sub-optimality."

---

> > ### Author Response · Authors · 2024-04-16
> > **Response**
> >
> > > I can roughly buy the justification for the result outlined in Remark 1, although the result would be stronger if the boundedness of iterates was guaranteed rather than assumed. However, it is not obvious to me why the statement starting with "the LHS is simply" is true. Does it not at least require assuming that has zero gradient at optimality? Writing out the identity explicitly would be useful. Moreover, presumably one should write $\nabla M_\theta \circ \nabla M_{\vartheta}^*$ instead of $\nabla M_\theta \nabla M_\vartheta^*$ since these are vector-valued functions and multiplying them doesn't make sense.
> >
> > Boundedness of the iterates could likely be enforced by assuming a coercivity-type condition on $f$ and performing careful analysis, but this does not add additional insights to the convergence. In our experiments, we did not find that the iterates diverged either when using the proposed dual parameterization.
> >
> > &nbsp;&nbsp;&nbsp;&nbsp;The LHS having the given form comes directly from expanding the definitions of $\tilde{z}^{(k+1)}$ and $\hat{z}^{(k+1)}$, as in Step 3 of Alg. 1 and Equation (5). We have corrected the formulations in Remark 1 to use $\circ$ for composition and have the iterates properly in the dual space, which was likely misleading.
> >
> > > It is not clear to me whether $\hat{z}^{(k)}$ refers to the iterates resulting from running "true" mirror descent for the entire trajectory or just the sequence resulting from applying the "true" mirror descent update to $\tilde{z}^{(k-1)}$. I believe it is the latter, but this should be made more clear (in particular, I'm not sure if they should be described as "iterates" since one does not follow from the other; it is instead an auxiliary sequence associated with the approximate mirror descent iterates $\tilde{z}^{(k)}$.
> >
> > $\hat{z}^{(k)}$ refers to the latter of the two mentioned, as in Equation (5), given by a true mirror descent iteration applied to $\tilde{z}^{(k-1)}$ (with $\nabla M_\theta$ being the forward map). We have changed the name from "iterates" to "auxiliary sequence" to clarify this.
> >
> > > I would suggest not using the word "regret" to describe any result in this paper or the results of Beck \& Teboulle (2003), since its meaning is defined using the setup in online learning rather than optimization. A better word might be "sub-optimality."
> >
> > We have changed instances of "regret" to "optimality gap".

---

### Review · Reviewer_QA8S · 2024-03-11

**Summary Of Contributions:**

This work aims to improve the optimization speed of the learned mirror descent method. This method is similar to classical mirror descent but uses a learnable mirror map (and its convex conjugate) parameterized by input-convex neural nets.

This work proposes the Nesterov-accelerated version, the stochastic version, and the stochastic accelerated version of learned mirror descent. And then it provides theoretical analyses showing that these methods converge to optimal values on convex functions, assuming certain error bounds on how well the learned maps are convex conjugate to each other.

Experiments are conducted on several convex optimization problems and neural network training.

**Audience:**

Yes

**Broader Impact Concerns:**

/

**Claims And Evidence:**

Yes

**Requested Changes:**

The theory part is good overall, but for the experiment part I hope the authors can resolve Weaknesses 1 and 2.

**Strengths And Weaknesses:**

Strengths:
1. This paper is well-written. The proof looks correct but I didn't have time to check all the details.
2. The assumptions are all reasonable and many are standard in the optimization literature.
3. The proposed methods are validated in various different settings to justify their effectiveness.

Weaknesses:
1. In the comparison with the GD, Nesterov and Adam baselines, the authors did not discuss how they picked the learning rates and whether they are thoroughly tuned with a grid search. If the learning rates are not tuned, then the current experiments may underestimate the optimization speed of the baselines.
2. The experiments only measure the number of iterations, but applying the learned mirror maps can lead to extra computation costs. It would be valuable if the authors could report the wall-clock time as well.
3. Experiments on neural net training illustrate the effectiveness of the proposed methods beyond convex optimization, but these experiments are limited to rather shallow neural nets trained with only a couple of epochs.

---

> ### Author Response · Authors · 2024-03-20
> **Rebuttal to Reviewer QA8S**
>
> We thank the reviewer for the kind feedback and suggestions. Please find below point-by-point responses to the reviewer's concerns.
>
> > In the comparison with the GD, Nesterov and Adam baselines, the authors did not discuss how they picked the learning rates and whether they are thoroughly tuned with a grid search. If the learning rates are not tuned, then the current experiments may underestimate the optimization speed of the baselines.
>
> We thank the reviewer for pointing this out, as the baselines were quickly tuned near the standard learning rates. The revised manuscript now uses learning rates tuned using a grid search. The convex experiments setting have been changed accordingly to incorporate this, and we have added a section in Appendix E detailing the effect of step size on the baseline methods, as well as the chosen step sizes for image denoising, inpainting, and SVM experiment. We note a trade-off between low-iteration convergence rate and high-iteration plateauing as seen in the newly added Figure 9 in Appendix E, which makes this a subjective choice. Note that even after fine-tuning the step size of the baselines using a grid search, they get outperformed by LAMD in terms of overall time (as seen in Figure 10).
>
> > The experiments only measure the number of iterations, but applying the learned mirror maps can lead to extra computation costs. It would be valuable if the authors could report the wall-clock time as well.
>
> We have added additional figures in Appendix F that plot the target loss against wall-clock time of the proposed methods against the baselines. This is done for the image inpainting and the CNN training experiments. We observe in Figure 10 that LAMD performs favorably for the image inpainting experiment, faster than the baseline methods in the medium to high iteration regime. Figure 11 demonstrates for CNN training using equivariance that the time increase of LMD/LAMD compared to the baselines is small despite a lack of optimization.
>
> > Experiments on neural net training illustrate the effectiveness of the proposed methods beyond convex optimization, but these experiments are limited to rather shallow neural nets trained with only a couple of epochs.
>
> We have added additional experiments on few-shot learning to demonstrate the prototypical application and convergence behavior for non-convex problems. Additional explanation of the baseline meta-learning method can be found in Appendix H.

---

### Review · Reviewer_Czv7 · 2024-03-12

**Summary Of Contributions:**

There is a growing body of literature on "learning-to-optimize" (L2O) methods. Broadly speaking, these methods use learning-based strategies to accelerate the solution of specific instances of optimization algorithms, given a training set of solution outputs from previously seen instances. This paper focuses on learning-based extensions of mirror descent (MD), which enables optimization over convex functions over non-Euclidean spaces (such as the simplex).

In previous work, Tan-Mukherjee-Tang-Hauptmann-Schonlieb (2023a,b) proposed and studied learned mirror descent (LMD). At a high level this takes the standard mirror descent algorithm of Nemirovski and Yudin (1983) and replaces the mirror potential (and its convex conjugate) with learned functions parameterized by a pair of neural networks. This work extends this earlier study from 2023 in a few different ways: fixing certain algorithmic inconsistencies in LMD; achieving further acceleration; analyzing stochastic variants; and reducing the sample complexity of learning.

Within this overall context, the authors make the following specific contributions:
* They propose and analyze learned accelerated mirror descent (LAMD), which fixes the *forward-backward* inconsistency that arises in LMD due to the fact that the "mirror map" neural network and the "conjugate" neural network are in fact not guaranteed to be convex conjugates of each other. In the previous work on LMD, the analysis swept this discrepancy "under the rug" using an inexact gradient oracle; here, the authors provide a more salient algorithmic fix. Theoretical analysis shows that under certain assumptions, LAMD gives a true minimum, as opposed to giving an additive error factor above the minimum.
* They propose and analyze stochastic versions of LMD and LAMD, where the gradients are supplied by noisy/stochastic oracles.
* They propose using *equivariance* to reduce the number of parameters required to learn the mirror maps. This arises due to the property that MD methods remain permutation invariant throughout training,

**Audience:**

Yes

**Claims And Evidence:**

No

**Requested Changes:**

* Consider restructuring the paper where the first 10-12 pages summarize the most important parts of the paper, and relegate many of the more repetitive parts to an appendix.
* The paragraph "We note that this analysis can be used to partially explain the existing heuristic..." is confusing. Can you explain more clearly what these earlier methods do, and how your analysis partially explains existing heuristics?
* More clarity on the monotonic splines experiment can be helpful -- why this is a meaningful choice of mirror map, what the spline order is, what the variables are etc. Maybe a figure, or clear equations, can help.

**Strengths And Weaknesses:**

**Strengths**

* Nice and extensive set of follow-up contributions to a specific method (LMD) in the L2O literature.
* Relevant to the TMLR audience.


**Weaknesses**

* The paper may benefit from restructuring. Right now, the algorithmic treatment is somewhat repetitive with little qualitative or analysis insights separating the various accelerated versions. Moving large chunks of Sections 2,3,4, to an appendix can make the paper much more readable and will have the added benefit of cutting down its length (currently, 31 full pages of text).
* The part about equivariance is a bit orthogonal to the central problem addressed by this paper (mirror descent and its accelerated variants), and feels tacked on rather than being germane to the overall discussion. In any case it seems that any benefits to acceleration are low, since the authors could only demonstrate results on very simple toy neural networks.
* I am not very convinced by the experiments on equivariance, and would appreciate it if the authors provided further quantitative evidence (beyond trying it out for MNIST).
* Related to the above two points: it is not entirely clear to me whether reducing the number of parameters via equivariance is a good idea. We do now know that overparameterization can be a benefit in various ways in learning deep neural networks -- either by making the optimization landscape more benign, or by adding additional implicit regularization to improve generalization. Weight tying may not always be a good thing. I would encourage the authors to think a bit more deeper into whether this section really makes the paper stronger.

---

> ### Author Response · Authors · 2024-03-20
> **Rebuttal to Reviewer Czv7**
>
> We thank the reviewer for the detailed and useful feedback. Please find below point-by-point responses to the reviewer's concerns.
>
> > The paper may benefit from restructuring. Right now, the algorithmic treatment is somewhat repetitive with little qualitative or analysis insights separating the various accelerated versions. Moving large chunks of Sections 2,3,4, to an appendix can make the paper much more readable and will have the added benefit of cutting down its length (currently, 31 full pages of text).
>
> We have reorganized the paper as suggested by the reviewer as well as Reviewer QBq1, moving many of the algorithmic proofs for the accelerated and stochastic portions (Sections 2-4) to the appendix. There is some explanation at the end of Sections 2-4 that relate the results to previous literature.
>
> > The part about equivariance is a bit orthogonal to the central problem addressed by this paper (mirror descent and its accelerated variants), and feels tacked on rather than being germane to the overall discussion. In any case it seems that any benefits to acceleration are low, since the authors could only demonstrate results on very simple toy neural networks.
>
> Related to point (4), see response below.
>
> > I am not very convinced by the experiments on equivariance, and would appreciate it if the authors provided further quantitative evidence (beyond trying it out for MNIST).
>
> As suggested by Reviewer QBq1 who shared the same concerns, we have added additional few-shot learning experiments on the Omniglot dataset, as well as some baselines in this setting.
>
> > Related to the above two points: it is not entirely clear to me whether reducing the number of parameters via equivariance is a good idea. We do now know that overparameterization can be a benefit in various ways in learning deep neural networks -- either by making the optimization landscape more benign, or by adding additional implicit regularization to improve generalization. Weight tying may not always be a good thing. I would encourage the authors to think a bit more deeper into whether this section really makes the paper stronger.
>
> We address the second point along with this point by clarifying the intended purpose of the equivariance section as a general computational method for L2O. This method is then applied to the LMD framework on a sensitive non-convex problem, by reducing the parameters sufficiently that an exact mirror map parameterization is feasible. The sensitivity of the neural network to its parameters leads to the failure of LMD due to the usage of approximate mirror maps, necessitating exact mirror maps for this use-case. The analysis presented is also only for the convex case. Equivariance in this case allows for reparameterization using exact mirror maps with a small amount of meta-parameters to allow for computational tractability.
>
> &nbsp;&nbsp;&nbsp;&nbsp;While overparameterization may indeed be useful in the supervised setting for better generalization, we note that the notion of generalization for LMD has subtle differences compared to classical generalization. In LMD, good generalization refers to fast optimization outside the trained number of iterations, which we believe has been sufficiently demonstrated. This is opposed to the classical notion of generalization, which in the context of LMD, would refer to fast convergence for the trained number of iterations $N=10$ on unseen problems in the same class. We note that this classical notion is shown in the experiments as we test the LMD methods for higher iteration counts. We refer to (Tan et al., 2023a) for results in the case where LMD is transferred from denoising to deconvolution.
>
> &nbsp;&nbsp;&nbsp;&nbsp;Of course, if the optimization method is initialized away from gradient descent (or is otherwise not equivariant), then the resulting L2O schemes may not be equivariant. We leave a more detailed analysis and possible improvements in the neural network setting as future work.
>
> ----
> > Consider restructuring the paper where the first 10-12 pages summarize the most important parts of the paper, and relegate many of the more repetitive parts to an appendix.
>
> The proofs have been moved to the appendix in the revised manuscript.

---

> > ### Author Response · Authors · 2024-03-20
> > **Followup to Rebuttal**
> >
> > > The paragraph "We note that this analysis can be used to partially explain the existing heuristic..." is confusing. Can you explain more clearly what these earlier methods do, and how your analysis partially explains existing heuristics?
> >
> > The referenced papers (D'Orazio et al., 2021), (Gunasekar et al., 2018) consider mirror maps where the parameters are taken to be in $\mathbb{R}^d$, where $\psi(w) = \|w\|_p^2/2,\, 1<p\le 2$. (Ajanthan et al., 2021) instead considers element-wise tanh-projection and softmax on the whole parameter space, both of which are equivariant under say index permutations of a given layer. We note that one MD method by (Gao et al., 2023) considers the alternative setting where the block-diagonal mirror map decouples layers, but allows gradients between features of a given layer to interact, generally falling outside the proposed equivariant theory. These maps can be thought of a special case of the equivariant framework as the mirror maps act equally on every element, which we generalize slightly by allowing for mirror maps to work on product parameter spaces $\mathbb{R}^{|\Theta|} = \mathbb{R}^{|W_1|} \times \mathbb{R}^{|W_2|} \times ... \times \mathbb{R}^{|W_L|}$, acting with some equivariance on each component.
> >
> > We have added part of this explanation to the main text in the revised version.
> >
> > > More clarity on the monotonic splines experiment can be helpful -- why this is a meaningful choice of mirror map, what the spline order is, what the variables are etc. Maybe a figure, or clear equations, can help.
> >
> > We have added an explanatory section in Appendix G regarding the monotonic spline, as well as a sentence pointing to the appendix in the main text.

---

### Decision · Action_Editor_zve8 · 2024-04-30

**Recommendation:** Accept as is

**Comment:**

The paper provides a correct and useful result as identified by the reviewers. The reviewers all voted leaning accept to accept. The paper meets all the relevant criteria for TMLR.

**Audience:**

Optimization researchers in the community will be interested in the paper.

**Claims And Evidence:**

The claims made in the paper were found to be accurate, convincing and the evidence was clearly provided according the reviewers.